# Differentially Private Fréchet Mean on the Manifold of Symmetric Positive Definite (SPD) Matrices with log-Euclidean Metric

**Saiteja Utpala**                                                                                     *saitejautpala@gmail.com*
*UC Santa Barbara*

**Praneeth Vepakomma**                                                                                 *vepakom@mit.edu*
*MIT*

**Nina Miolane**                                                                                       *ninamiolane@ucsb.edu*
*UC Santa Barbara*

**Reviewed on OpenReview:** *https://openreview.net/forum?id=mAx8QqZ14f*

## Abstract

Differential privacy has become crucial in the real-world deployment of statistical and machine learning algorithms with rigorous privacy guarantees. The earliest statistical queries, for which differential privacy mechanisms have been developed, were for the release of the sample mean. In Geometric Statistics, the sample Fréchet mean represents one of the most fundamental statistical summaries, as it generalizes the sample mean for data belonging to nonlinear manifolds. In that spirit, the only geometric statistical query for which a differential privacy mechanism has been developed, so far, is for the release of the sample Fréchet mean: the *Riemannian Laplace mechanism* was recently proposed to privatize the Fréchet mean on complete Riemannian manifolds. In many fields, the manifold of Symmetric Positive Definite (SPD) matrices is used to model data spaces, including in medical imaging where privacy requirements are key. We propose a novel, simple and fast mechanism - the *tangent Gaussian mechanism* - to compute a differentially private Fréchet mean on the SPD manifold endowed with the log-Euclidean Riemannian metric. We show that our new mechanism has significantly better utility and is computationally efficient — as confirmed by extensive experiments.

## 1 Introduction

Privacy-preserving computing is an active area of research which is necessitated by ethics, regulations, requirements for protections of trade secrets, or possible lack of trust amongst distributed data siloes. Privacy preservation is desired across several topologies of data sharing, be it from client devices to powerful centralized entities or a in peer-to-peer fashion. Mistrust in data sharing carries over not only in the sharing of raw data but also in the sharing of results obtained from intermediate or complete computations. The need for stringent privacy protections is often fueled by many privacy leakages and attacks that continue to happen under various settings operating without the right level of privacy-protecting mechanisms.

In this context, differential privacy (DP) (Dwork et al., 2006; Dwork, 2008; Dwork et al., 2014; Dwork, 2006) has emerged as one of the leading mathematical definitions to ensure the preservation of privacy up to a chosen level. Privacy-preserving *mechanisms* that satisfy the definition of differential privacy were subsequently developed to privatize a wide range of statistical and machine learning computations. The earliest queries, for which mechanisms have been proposed, were for the privatization of sample means in statistics, computed for data lying on linear spaces. When data belong to nonlinear manifolds, the

Fréchet mean query (Fréchet, 1948) is the foundational building block of geometric statistics that needs to be privatized. Our work proposes a new, simpler and faster, mechanism for private Fréchet means on the manifold of symmetric positive definite (SPD) matrices endowed with log-Euclidean metric.

## 1.1 Motivation

**Fréchet mean: a building block in geometric statistics** While traditional statistics studies data that lies on *linear spaces*, geometric statistics studies data that lies on *nonlinear spaces* such as Riemannian manifolds, affine connection spaces, or stratified spaces (Pennec et al., 2019; Miolane, 2016). Such analysis is fruitful as data might have inherent constraints that are well captured by the geometry of a nonlinear space Miolane et al. (2021); Myers et al. (2022). For instance, symmetric matrices constrained to have strictly positive eigenvalues are conveniently modeled as elements of the manifold of symmetric positive definite (SPD) matrices. Several extensions of traditional statistical analysis tools have thus been developed for the manifold setting: regression has been generalized to geodesic regression (Fletcher, 2011; Thomas Fletcher, 2013), principal component analysis (PCA) to principal geodesic analysis or geodesic PCA (Fletcher et al., 2004; Sommer et al., 2010; Huckemann et al., 2010), and mean shift to Riemannian mean shift clustering (Subbarao & Meer, 2009; Caseiro et al., 2012). In each of these algorithms, the computation of the *sample Fréchet mean* generalizes the computation of the *sample mean*, and thus represents the most fundamental building block. The privatization of the Fréchet mean is therefore the key element required to privatize geometric statistical queries. Privacy-preserving geometric statistics is also crucial, as one of its main application areas is medical imaging and computational anatomy (Pennec et al., 2019; Miolane, 2016) for which privacy requirements are often desirable.

**Importance of the SPD manifold with log-Euclidean metric** Symmetric positive definite (SPD) matrices model a wide range of data, from medical images with Diffusion Tensor Imaging (DTI) (Basser et al.; Pennec et al., 2006), to physiological signals with electroencephalography (EEG) signals from brain-computer interfaces (BCI)(Yger et al., 2016; Zanini et al., 2017; Chevallier et al., 2021), to 3D shapes (Tabia et al., 2014) to name a few. Given their central roles for medical data where privacy is of the utmost importance (Lotan et al., 2020; Li et al., 2005), private statistical computations on the SPD manifold are a worthy endeavour. The SPD manifold can be equipped with different *Riemannian metrics* that provide elementary operations such as distance computations. The log-Euclidean metric, originally proposed in (Arsigny et al., 2006), has numerous advantages over another popular Riemannian metric called the affine invariant metric (Pennec et al., 2006): (*a*) it is computationally faster, (*b*) it gives similar or better performances on several processing and learning tasks, (*c*) and quite importantly, it provides a *closed form* expression for the Fréchet mean - which otherwise requires solving an optimization problem.

**Need for better and faster privacy mechanisms** Despite its importance for the processing of a number of (medical) data, geometric statistics currently stands understudied from the lens of differential privacy. The very recent work by (Reimherr et al., 2021) provides the first differentially private mechanism for the Fréchet mean. However, its utility - a measure of the mechanism's deviation from non-privatized computations - makes it impracticable on the manifold of SPD matrices as soon as we consider matrices of moderate size, e.g. $20 \times 20$ matrices. Consequently, there is a need for better and faster privacy mechanisms on manifolds, starting with the SPD manifold.

## 1.2 Related Work and Contributions

Reimherr et al. (2021) were first to consider differential privacy in manifold setting and developed *Riemannian Laplace mechanism* by extending the standard Laplace mechanism (Dwork et al., 2014) for linear spaces to complete Riemannian manifolds. It is based on a Laplace distribution that was originally proposed for SPD matrices (Hajri et al., 2016) based on distance of the affine invariant metric (Pennec et al., 2006), which they generalize to any manifold $\mathcal{M}$ equipped with a distance $\rho$:

$$p(x) \propto \exp\left(-\frac{\rho(x, m)}{\sigma}\right), \quad \forall x \in \mathcal{M} \tag{1}$$

where $m \in \mathcal{M}$, $\sigma \in \mathbb{R}_{>0}$(positive reals) are parameters of the probability density $p$. Reimherr et al. (2021) show that the mechanism obtained achieves *pure* differential privacy and provides an upper bound for the expectation of its utility (a measure of the deviation from non-privatized computations) for the Fréchet mean query. Their method is applicable to various Riemannian manifolds that satisfy some regularity conditions.

Approximate differential privacy relaxes pure differential privacy (see Section 2) but provides significantly better utility for higher dimensions and is heavily used in real world applications Abadi et al. (2016). In the Euclidean case, the Gaussian mechanism, where noise is added from standard Gaussian, satisfies approximate differential privacy. To this end, we make use of log Gaussian distribution Schwartzman (2016), an intrinsic distribution on SPD matrices, for deriving approximate differentially private mechanism. This relaxation helps us obtain better utility compared to Riemannian Laplace mechanism in terms of dimension, similar to standard Euclidean case. We summarize our contributions are as follows.

| Mechanism $\mathbf{A}$ | DP | $\mathbb{E}[\rho^2(f(\mathcal{D}), \mathbf{A}(\mathcal{D}))]$ | Theoretical Results |
|---|---|---|---|
| Riemannian Laplace (Reimherr et al., 2021) | Pure DP | $\mathcal{O}(k^4)$ | Expectation of $\rho^2(f(\mathcal{D}), \mathbf{A}(\mathcal{D}))$ |
| tangent Gaussian (Ours) | Approx. DP | $\mathcal{O}(\ln(1/\delta)k^2)$ | Exact Distribution of $\rho^2(f(\mathcal{D}), \mathbf{A}(\mathcal{D}))$ |

Table 1: Differences between existing (Reimherr et al., 2021) and proposed mechanisms for private Fréchet mean queries on the manifold of $k \times k$ SPD matrices endowed with the log-Euclidean metric. The notation $\rho^2(f(\mathcal{D}), \mathbf{A}(\mathcal{D}))$ represents the utility with $\mathcal{D}$ the dataset, $\mathbf{A}$ the mechanism under consideration, $\rho$ the log-Euclidean distance, $f$ the Fréchet mean and $\delta$ quantifies approximate differential privacy.

1. We propose a new and simple mechanism - called the *tangent Gaussian Mechanism* - that privatizes any statistical summary on the manifold of Symmetric Positive Definite (SPD) matrices endowed with the log-Euclidean metric. We prove that it achieves approximate differential privacy (Th. 2).

2. When the statistical summary is the Fréchet mean, we show that our mechanism obtains significant improvement in terms of utility over recent works - which we demonstrate theoretically, and practically for data in higher dimensions. Further, our mechanism is computationally efficient and easily implementable.

3. We present the effectiveness of our mechanism on synthetic and real-world (medical) imaging data, the latter being represented via their covariance descriptors. To this aim, we also prove a theoretical bound on the radius of log-Euclidean geodesic ball with the covariance descriptor pipeline (Tuzel et al., 2006) - required for the applicability of our mechanism (Th. 5).

Table 1 highlights the technical differences between (Reimherr et al., 2021) and our work.

## 2 Preliminaries and Notations

**Elements of Riemannian Geometry** Let $\mathcal{M}$ be a $d$-dimensional smooth connected manifold and $T_p\mathcal{M}$ be its tangent space at point $p \in \mathcal{M}$. A *Riemannian metric $g$* on $M$ is a collection of inner products $g_p : T_p\mathcal{M} \times T_p\mathcal{M} \to \mathbb{R}$ that vary smoothly with $p$. A manifold $\mathcal{M}$ equipped with a Riemannian metric $g$ is called a Riemannian manifold. Importantly, the metric $g$ gives a distance $\rho$ on $\mathcal{M}$. Let $\gamma : [0, 1] \to \mathcal{M}$ be a smooth parametrized curve on $\mathcal{M}$ with velocity vector at $t$ denoted as $\dot{\gamma}_t \in T_{\gamma(t)}\mathcal{M}$. The length of $\gamma$ is defined as $L_\gamma = \int_0^1 \sqrt{g_{\gamma(t)}(\dot{\gamma}_t, \dot{\gamma}_t)}dt$ and the distance $\rho$ between any two points $p, q \in \mathcal{M}$ is: $\rho(p, q) = \inf_{\gamma:\gamma(0)=p,\gamma(1)=q} L_\gamma$.

If in addition $\mathcal{M}$ is complete for $\rho$, then any two points $p, q \in \mathcal{M}$ can be joined by length-minimizing curve, called a geodesic. We refer the reader to (Do Carmo & Flaherty Francis, 1992; Lee, 2006; Helgason, 1979) for a detailed exposition.

**Elements of Differential Privacy (DP)** Let $\mathcal{X}$ be an input data space and $\mathcal{M}$ the manifold under consideration. Let $f : \mathcal{X}^n \to \mathcal{M}$ be a manifold-valued statistical summary that requires privatization with respect to some sensitive dataset $\mathcal{D}$ of size $n$, *i.e.* $\mathcal{D} \in \mathcal{X}^n$. Two datasets $\mathcal{D}, \mathcal{D}' \in \mathcal{X}^n$ are said to be adjacent

if they differ by at most one data point. We denote adjacency as $\mathcal{D} \sim \mathcal{D}'$. The *sensitivity* of the summary $f$ with respect to the distance $\rho$ on $\mathcal{M}$ is defined as:

$$\Delta_\rho = \sup_{\mathcal{D} \sim \mathcal{D}'} \rho(f(\mathcal{D}), f(\mathcal{D}')), \tag{2}$$

which is the maximum amount of deviation that can occur in the output of $f$ for adjacent datasets.

A *mechanism* $\mathbf{A} : \mathcal{X}^n \to \mathcal{M}$ is a randomized algorithm that takes a dataset $\mathcal{D}$ as input, and outputs a privatized version of the summary $f$ on $\mathcal{D}$. The mechanism $\mathbf{A}$ satisfies $(\epsilon, 0)$ differential privacy (also *pure differential privacy*) if, for all adjacent datasets $\mathcal{D} \sim \mathcal{D}'$ and for all measurable sets $S$ of $\mathcal{M}$ the following holds:

$$\mathbb{P}[\mathbf{A}(\mathcal{D}) \in S] \leq \exp(\epsilon) \mathbb{P}[\mathbf{A}(\mathcal{D}') \in S] \tag{3}$$

The intuition is that the change of a single element of the data space $\mathcal{X}$ does not significantly alter the output distribution of the mechanism. As a relaxation, the mechanism $\mathbf{A}$ satisfies $(\epsilon, \delta)$-differential privacy (also *approximate differential privacy*) if, for all adjacent datasets $\mathcal{D} \sim \mathcal{D}'$ and for all measurable sets $S$ of $\mathcal{M}$:

$$\mathbb{P}[\mathbf{A}(\mathcal{D}) \in S] \leq \exp(\epsilon) \mathbb{P}[\mathbf{A}(\mathcal{D}') \in S] + \delta.$$

Intuitively, $\delta$ can be thought of as the probability of privacy failure, when Eq. equation 3 is not guaranteed.

Let $p_{\mathbf{A}(\mathcal{D})}$ be the density of the random variable $Y = \mathbf{A}(\mathcal{D})$. Given adjacent datasets $\mathcal{D} \sim \mathcal{D}'$, the *privacy loss function* of $\mathbf{A}$ is defined as

$$\ell_{\mathbf{A}, \mathcal{D}, \mathcal{D}'}(y) = \ln\left(\frac{p_{\mathbf{A}(\mathcal{D})}(y)}{p_{\mathbf{A}(\mathcal{D}')}(y)}\right) \quad \forall y \in \mathcal{M}, \tag{4}$$

and the *privacy loss random variable* is $L_{\mathbf{A}, \mathcal{D}, \mathcal{D}'} = \ell_{\mathbf{A}, \mathcal{D}, \mathcal{D}'}(Y)$ (Balle & Wang, 2018). Importantly for our derivations, both sufficient and sufficient & necessary conditions for the mechanism $\mathbf{A}$ to be $(\epsilon, \delta)$-differentially private (DP) can be formulated in terms of $L_{\mathbf{A}, \mathcal{D}, \mathcal{D}'}$. The sufficient condition writes : $\forall \mathcal{D} \sim \mathcal{D}' : \mathbb{P}[L_{\mathbf{A}, \mathcal{D}, \mathcal{D}'} \geq \epsilon] \leq \delta \implies \mathbf{A}$ is$(\epsilon, \delta)$-DP. The sufficient & necessary condition is: $\forall \mathcal{D} \sim \mathcal{D}' : \mathbb{P}[L_{\mathbf{A}, \mathcal{D}, \mathcal{D}'} \geq \epsilon] - \exp(\epsilon)\mathbb{P}[L_{\mathbf{A}, \mathcal{D}, \mathcal{D}'} \leq -\epsilon] \leq \delta \iff \mathbf{A}$ is$(\epsilon, \delta)$-DP.

**Fréchet Mean** When the data space $\mathcal{X}$ is equal to the manifold $\mathcal{M}$, we will be interested in mechanisms that can privatize a specific statistical summary $f$ called the Fréchet mean. The sample Fréchet mean $\overline{X}$ (Fréchet, 1948) of the dataset $\mathcal{D} = \{X_1, \dots X_n\}$ on the manifold $\mathcal{M}$ is defined as

$$\overline{X} \triangleq \left\{ p | p \in \arg\min_{q \in \mathcal{M}} \sum_{i=1}^{n} \rho^2(q, X_i) \right\},$$

*i.e.* we have in this case $\overline{X} = f(\mathcal{D})$ for $\mathcal{D} \in \mathcal{M}^n$. Intuitively, the Fréchet mean uses a property of the mean on linear spaces - namely the fact that mean minimizes the sum of squared distances to the data points - as a definition of mean on manifolds. Crucially, the Fréchet mean depends on the distance $\rho$ and therefore on the Riemannian metric defined on $\mathcal{M}$. We also note that the Fréchet mean might not always exist, and if it exists it might not be unique – see supplementary materials. In practice, computing $\overline{X}$ generally requires optimization algorithms such as gradient descent on manifolds (Boumal, 2020).

## 3 Geometry of the SPD Manifold with Log Euclidean Metric

**Manifold and vector space structures** We now restrict $\mathcal{M}$ to be the manifold of symmetric positive definite (SPD) matrices:

$$\mathrm{SPD}(k) = \left\{ X \in \mathbb{R}^{k \times k} | X^T = X \text{ and } \forall u \in \mathbb{R}^k \setminus \{0\}, u^T X u > 0 \right\}, \tag{5}$$

which has dimension $d = \frac{k(k+1)}{2}$. The tangent space of the manifold $\mathrm{SPD}(k)$ at any point $X \in \mathrm{SPD}(k)$ is the vector space of symmetric matrices $\mathrm{SYM}(k)$. The mathematical construct $(\mathrm{SPD}(k), +, .)$ is not a vector space

under element-wise addition and element-wise scalar multiplication. This can be seen from the observation that $a \in \mathbb{R}_{\leq 0}, X \in \text{SPD}(k) \implies aX \notin \text{SPD}(k)$. Instead, $\text{SPD}(k)$ is an open cone of $\mathbb{R}^{k \times k}$ and, as such, naturally possesses a smooth manifold structure which can further be equipped with different Riemannian metrics (Thanwerdas & Pennec, 2021). However, Arsigny *et al.* (Arsigny et al., 2007) showed in a surprising result that $\text{SPD}(k)$ can be given a vector space structure $(\text{SPD}(k), \oplus, \odot)$ via the operations $\oplus, \odot$ defined in Table 2, where Expm, Logm denote the matrix exponential and matrix logarithm. This fact is central for the proofs provided in the present paper.

| Operation | Notation | Expression |
|---|---|---|
| Addition | $X_1 \oplus X_2$ | $\text{Expm}\left[\text{Logm}\, X_1 + \text{Logm}\, X_2\right]$ |
| Subtraction | $X_1 \ominus X_2$ | $\text{Expm}\left[\text{Logm}\, X_1 - \text{Logm}\, X_2\right]$ |
| Scalar Multiplication | $a \odot X$ | $\text{Expm}\left[a.\,\text{Logm}\, X\right]$ |

Table 2: Operations turning the manifold $\text{SPD}(k)$ into a vector space. Expm and Logm denote the matrix exponential and logarithms, respectively. $X_1, X_2$ belong to $\text{SPD}(k)$ while $a \in \mathbb{R}$ is a scalar.

**Riemannian structure** Arsigny *et al.* further define a Riemannian metric on $\text{SPD}(k)$, called the *log-Euclidean metric*, which induces the following distance:

$$\rho_{\text{LE}}(X_1, X_2) = \|\text{Logm}\, X_1 - \text{Logm}\, X_2\|_F, \quad \forall X_1, X_2 \in \text{SPD}(k), \tag{6}$$

where $\|.\|_F$ denotes the Frobenius norm on matrices. Importantly, the log-Euclidean metric (Arsigny et al., 2006) gives a unique and simple closed form expression for the Fréchet mean in terms of matrix logarithm and matrix exponential

$$\overline{X}_{\text{LE}} = \text{Expm}\left[\frac{1}{n} \sum_{i=1}^{n} \text{Logm}\, X_i\right],$$

for the dataset $X_1, ..., X_n \in \text{SPD}(k)$.

**Maps between spaces** Lastly, we present maps that will help us define the differential privacy mechanism proposed in the next section. Consider the map $\text{vecd} : \text{SYM}(k) \to \mathbb{R}^{\frac{k(k+1)}{2}}$ defined as $\text{vecd}(X) = \left[\text{diag}(X)^T, \sqrt{2}\, \text{upperdiag}(X)^T\right]^T$, where $\text{diag} : \text{SYM}(k) \to \mathbb{R}^k$ and $\text{upperdiag} : \text{SYM}(k) \to \mathbb{R}^{\frac{k(k-1)}{2}}$ build vectors from the diagonal, and from the strictly upper diagonal entries, of the matrix $X$. The map vecd is invertible and we denote by invvecd its inverse. Specifically, the spaces $\text{SPD}(k), \text{SYM}(k)$ and $\mathbb{R}^{\frac{k(k+1)}{2}}$ are now related as follows:

$$\text{SPD}(k) \xrightleftharpoons[\text{Expm}]{\text{Logm}} \text{SYM}(k) \xrightleftharpoons[\text{invvecd}]{\text{vecd}} \mathbb{R}^{\frac{k(k+1)}{2}}.$$

## 4 Tangent Gaussian Mechanism on SPD manifolds

We can now introduce our differential privacy mechanism for statistical summaries on the $\text{SPD}(k)$ manifold. Let $f : \mathcal{X}^n \to \text{SPD}(k)$ be any $\text{SPD}(k)$-valued summary that needs to be privatized. The proposed mechanism is based on the log Gaussian distribution on the SPD manifold (Schwartzman, 2016) which is defined as follows. Consider a mean $M \in \text{SPD}(k)$ and a tangent covariance $\Sigma \in \text{SPD}\left(\frac{k(k+1)}{2}\right)$. We can (i) first map the mean $M$ to the tangent space $\text{SYM}(k)$ of $\text{SPD}(k)$ at the identity using the matrix Logarithm Logm, then (ii) to $\mathbb{R}^{\frac{k(k+1)}{2}}$ using the map vecd introduced in the previous section, and (iii) consider whether the result follows a traditional Gaussian distribution.

**Definition 1** (Log Gaussian Distribution on $\text{SPD}(k)$ (Schwartzman, 2016))**.** *Given a mean $M \in \text{SPD}(k)$, and a tangent covariance $\Sigma \in \text{SPD}\left(\frac{k(k+1)}{2}\right)$, we say that $X \sim \mathcal{LN}(M, \Sigma)$ follows a log Gaussian distribution*

---

**Algorithm 1:** tangent Gaussian Mechanism for $f : \mathcal{X}^n \to \text{SPD}(k)$

---

**Inputs :** Dataset $\mathcal{D}$ of $k \times k$ SPD matrices of size $n$, sigma-type $\in$ {'classical','analytic' }, $\Delta_{\text{LE}}$ the log-Euclidean sensitivity of $f$, $\epsilon > 0$, $\delta \in (0,1)$ and additionally $\epsilon < 1$ if sigma-type is 'classical', the noise calibration subroutines CLASSIC, ANALYTIC which take $\Delta_{LE}, \epsilon, \delta$ and provide $\sigma$.

**Output:** Private $f(\mathcal{D})$

**1** **if** sigma-type is 'classical' **then** $\sigma = \text{CLASSIC}(\Delta_{\text{LE}}, \epsilon, \delta)$; **else** $\sigma = \text{ANALYTIC}(\Delta_{\text{LE}}, \epsilon, \delta)$;

**2** Compute non private output : $f_{\text{np}} := f(\mathcal{D})$

**3** Compute mean of Gaussian distribution: $M := \text{vecd}[\text{Logm} \, f_{\text{np}}]$, $M \in \mathbb{R}^{\frac{k(k+1)}{2}}$

**4** Sample from the Gaussian distribution in $\mathbb{R}^{\frac{k(k+1)}{2}}$: $N \sim \mathcal{N}(M, \sigma^2 I)$

**5** Map sample to the SPD manifold: $f_p := \text{Expm}[\text{invvecd} \, N]$

**6** Return private $f_p$

---

*on* $\text{SPD}(k)$ *if* $\text{vecd}[\text{Logm} \, X] \sim \mathcal{N}(\text{vecd}[\text{Logm} \, M], \Sigma)$ *follows a (regular) Gaussian distribution with mean* $\text{vecd} \, \text{Logm} \, M$ *and covariance matrix* $\Sigma$ *on* $\mathbb{R}^{\frac{k(k+1)}{2}}$.

*The density* $p(X|M, \Sigma)$ *is then given by*

$$\frac{J(X)}{(2\pi)^{\frac{d}{2}} (\det \Sigma)^{\frac{1}{2}}} \exp\left(-\frac{1}{2} \text{vecd}(\text{Logm} \, X - \text{Logm} \, M)^T \Sigma^{-1} \text{vecd}(\text{Logm} \, X - \text{Logm} \, M)\right)$$

*where* $d = \frac{k(k+1)}{2}$, $J(X) = \frac{1}{\det X} \prod_{i<j} h(\lambda_i, \lambda_j)$, *and* $h(\lambda_i, \lambda_j) = \begin{cases} (\log \lambda_i - \log \lambda_j) & \lambda_i > \lambda_j \\ \frac{1}{\lambda_i} & \lambda_i = \lambda_j \end{cases}$, *with* $\lambda_i, \lambda_j$ *eigenvalues of the matrix* $X$.

The definition of log Gaussian distribution on the $\text{SPD}(k)$ manifold allows us to define our proposed tangent Gaussian mechanism.

**Definition 2** (tangent Gaussian Mechanism). *Consider any statistical summary* $f : \mathcal{X}^n \to \text{SPD}(k)$ *on the manifold* $\text{SPD}(k)$ *equipped with log-Euclidean metric. Given* $\sigma^2 > 0$, *we define the tangent Gaussian mechanism* $\mathbf{A}_{\text{TG}} : \mathcal{X}^n \to \text{SPD}(k)$, *as*

$$\mathbf{A}_{\text{TG}}(\mathcal{D}) = X, where \, X \sim \mathcal{LN}(f(\mathcal{D}), \sigma^2 I).$$

We now state our main theorem, which shows that the privacy loss of the tangent Gaussian mechanism is normally distributed with mean and variance parametrized by the log-Euclidean distance. Proof is given in Appendix A.2

**Theorem 1** (Distribution of Privacy Loss for the tangent Gaussian Mechanism). *Let* $\mathbf{A}_{TG}$ *be a tangent Gaussian mechanism with variance* $\sigma^2$. *Its privacy loss is normally distributed as*

$$L_{\mathbf{A}_{\text{TG}}, \mathcal{D}, \mathcal{D}'} \sim \mathcal{N}\left(\frac{\rho_{\text{LE}}^2(f(\mathcal{D}), f(\mathcal{D}'))}{2\sigma^2}, \frac{\rho_{\text{LE}}^2(f(\mathcal{D}), f(\mathcal{D}'))}{\sigma^2}\right).$$

This distribution is analogous to the distribution of the privacy loss for the Euclidean Gaussian mechanism, but with the log-Euclidean sensitivity instead of the Euclidean sensitivity (Dwork et al., 2014; Balle & Wang, 2018). Consequently, our theoretical analysis of the tangent Gaussian mechanism - deriving privacy guarantees from the distribution of the privacy loss above - closely follows the steps of the analysis for the Euclidean Gaussian case. Specifically, we can proceed in two ways with either a (1) classical approach where sufficient conditions are used to show the mechanism is $(\epsilon, \delta)$-DP as in Dwork et al. (2014), or with an (2) analytic approach where the utility is better by using sufficient and necessary conditions (Balle & Wang, 2018).

**Theorem 2** (Privacy Guarantee of tangent Gaussian Mechanism). *Consider* $f : \mathcal{X}^n \to \text{SPD}(k)$ *with log-Euclidean sensitivity* $\Delta_{\text{LE}}$.

1. *(Classical) Given $\epsilon, \delta \in (0,1)$, choosing $\sigma = \Delta_{\text{LE}}\sqrt{2\ln(1.25/\delta)}/\epsilon$, makes the tangent Gaussian mechanism $(\epsilon, \delta)$-differentially private.*

2. *(Analytic) Given $\epsilon \geq 0, \delta \in (0,1)$ and $\Phi$ the cumulative distribution of the standard Gaussian, choosing any $\sigma$ that satisfies $\Phi\left(\frac{\Delta_{\text{LE}}}{2\sigma} - \frac{\epsilon\sigma}{\Delta_{\text{LE}}}\right) - \exp(\epsilon)\Phi\left(\frac{\Delta_{\text{LE}}}{2\sigma} - \frac{\epsilon\sigma}{\Delta_{\text{LE}}}\right) \leq \delta$ makes the tangent Gaussian mechanism $(\epsilon, \delta)$-differentially private.*

Proofs are is given in Appendix A.3. Algorithm 1 shows the implementation of the mechanism.

## 5   Privatizing the Fréchet mean

In the previous section, $f$ is any function that outputs a summary statistics on $\text{SPD}(k)$. In this section, we seek to privatize the Fréchet mean $f$ of the log-Euclidean metric. We first compute its sensitivity and then provide its utility. In what follows, $\mathcal{B}_r(M) = \{X | \rho_{\text{LE}}(M, X) < r\}$ denotes an open geodesic ball of radius $0 < r < \infty$ centered at $M \in \text{SPD}(k)$.

**Theorem 3** (Sensitivity of Log-Euclidean Fréchet Mean)**.** *Given data in $\mathcal{B}_r(M)$ for some $0 < r < \infty$ and $M \in \text{SPD}(k)$, the sensitivity of the log-Euclidean Fréchet mean verifies: $\Delta_{\text{LE}} \leq \frac{2r}{n}$.*

Note above theorem can also obtained from (Reimherr et al., 2021, Theorem 2) by setting $\kappa = 0$. The utility of the tangent Gaussian mechanism for a Fréchet mean query is then given below.

**Theorem 4** (Utility)**.** *Let $\mathbf{A}_{\text{TG}}$ be the (classical) tangent Gaussian mechanism, $\mathcal{B}_r(M)$ a geodesic ball of radius $0 < r < \infty$ and center $M \in \mathcal{M}$ containing the dataset $\mathcal{D}$ and $f$ the Fréchet mean. The utility of the mechanism $\mathbf{A}_{\text{TG}}$ is given by:*

$$\rho_{\text{LE}}^2(f(\mathcal{D}), \mathbf{A}_{\text{TG}}(\mathcal{D}))) \sim \sigma^2 \chi_d^2,$$
$$\mathbb{E}[\rho_{\text{LE}}^2(f(\mathcal{D}), \mathbf{A}_{\text{TG}}(\mathcal{D}))] = \frac{4r^2\ln(1.25/\delta)d}{n^2\epsilon^2} \qquad \text{with } d = dim(\text{SPD}(k)) = \frac{k(k+1)}{2},$$

*where $\chi_d^2$ represents the chi squared distribution with $d$ degree of freedoms.*

Proofs of Th. 3 and Th. 4 are given in Appendix A.4. We compare these results with those of the Riemannian Laplace mechanism (Reimherr et al., 2021), denoted $\mathbf{A}_{\text{RL}}$.

**Utility**: We compare the utility in terms of size $k$ of spd matrices $k \times k$ because dependancy on other factors $n, \epsilon$ are same. Utility of the Riemannian Laplace mechanism has an expectation given by $\mathbb{E}[\rho_{\text{LE}}^2(f(\mathcal{D}), \mathbf{A}_{\text{RL}}(\mathcal{D}))] = \mathcal{O}(k^4)$. By contrast, our tangent Gaussian mechanism provides $\mathbb{E}[\rho_{\text{LE}}^2(f(\mathcal{D}), \mathbf{A}_{\text{TG}}(\mathcal{D}))] = \mathcal{O}(\ln(1/\delta)k^2)$. Hence our mechanism has significantly better utility in terms of dimension.

**Pure DP vs Approx DP**: It should be noted that our privacy guarantees are weaker than Riemannian Laplace. In practice, $\delta$ is chosen to be cryptographically small and typically $\delta \ll 1/n$ Canonne (2021).

**Theoretical Results**: The authors of Reimherr et al. (2021) characterize the utility in terms of its expectation $\mathbb{E}[\rho_{\text{LE}}^2(f(\mathcal{D}), \mathbf{A}(\mathcal{D}))]$. By contrast, our results yield a more complete picture, as we derive the probability distribution of $\rho_{\text{LE}}^2(f(\mathcal{D}), \mathbf{A}(\mathcal{D})))$ given that we are tailoring mechanism for flat geometry of SPD matrices with log-Euclidean metric.

## 6   Experiments

We use the Riemannian Laplace mechanism as the baseline and recall that this mechanism uses the Riemannian Laplace distribution equation 25. Efficient sampling from the Riemannian Laplace distribution is only discussed for $(i)$ SPDManifold with affine-invariant metric and $(ii)$ Hypersphere with Euclidean metric in Reimherr et al. (2021) and we didn't find any sampling procedure from this distribution on SPD manifold with log-Euclidean metric in Reimherr et al. (2021); Hajri et al. (2016) and hence we used MCMC sampling in our experiments.

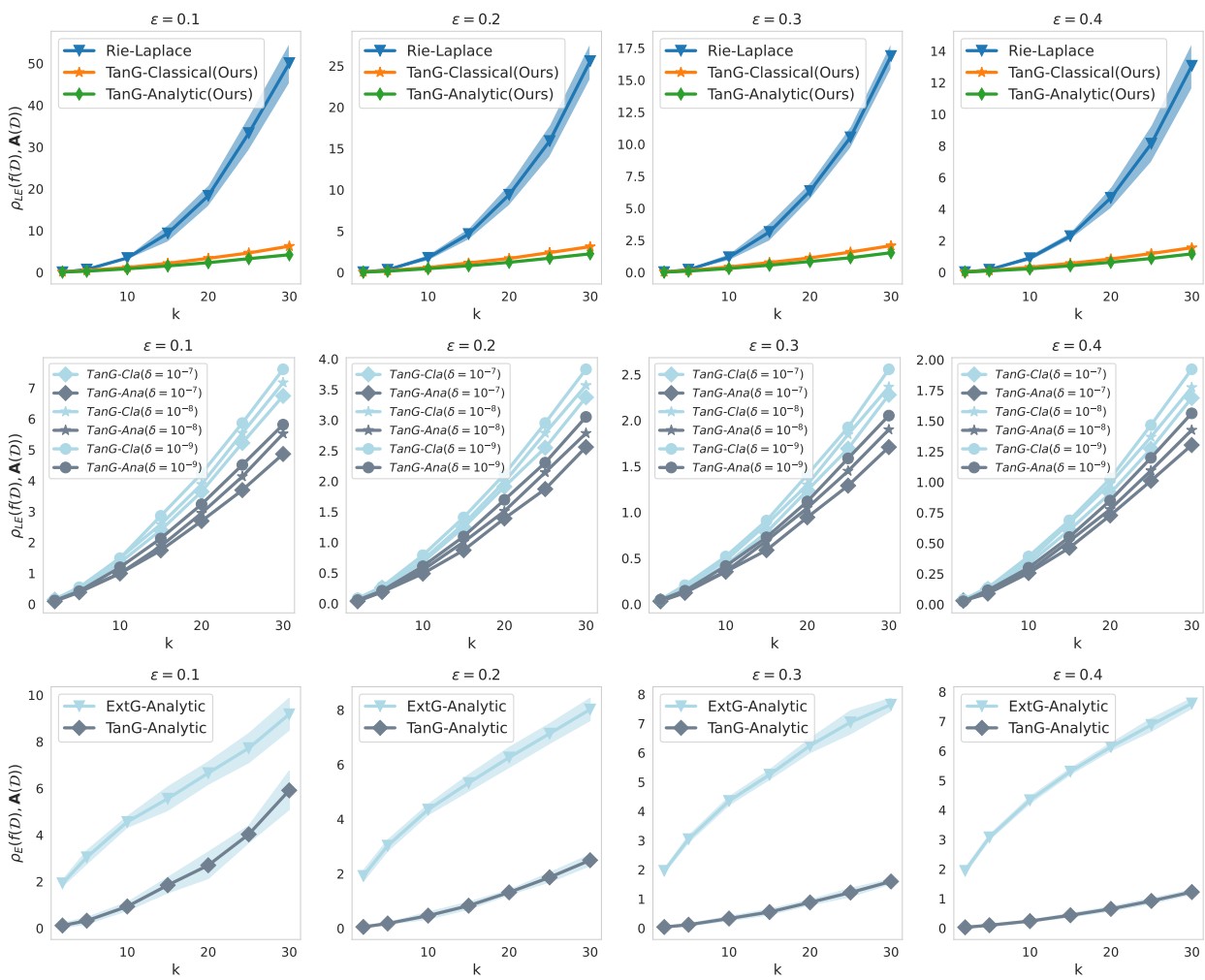

Figure 1: Utilities on synthetic data for *Rie-Laplace* the Riemannian Laplace mechanism (Reimherr et al., 2021), and *TanG Classical, TanG-Cla* and *TanG Analytic, TanG-Ana* our proposed tangent Gaussian mechanisms (classical and analytic versions), and *ExtG-Analytic* the Extrinsic analytic gaussian mechanism for different matrix sizes $k$ and privacy parameter $\epsilon$. $\rho_{LE}$ and $\rho_E$ denotes log-Euclidean and Euclidean distance respectively. Note output of extrinsic mechanism is not a SPD matrix and hence deviation is measured in standard Euclidean distance.

## 6.1 Experiments on Synthetic Datasets

The utility depends on privacy parameters $(\epsilon, \delta)$, the size $k$ of the matrices, the dataset size $n$ and $r$ the radius of the geodesic ball containing the dataset. The utilities of the tangent Gaussian and Riemannian Laplace mechanisms have the same dependency on $n, \epsilon, r$, such that their differentiating parameters are $\delta, k$. Consequently, our experiments on synthetic data fix $n, \epsilon, r$ and vary $\delta, k$.

We also consider Extrinsic approach suggested in Reimherr et al. (2021) where Fréchet mean is seen to be belonging to Symmetric matrix and noise from Euclidean normal distribution is added, specifically $\mathbf{A}_{\mathrm{EX}}(\mathcal{D}) = X, X \sim \mathrm{invvecd}\left(\mathcal{N}\left(\mathrm{vecd}\,\bar{\mathrm{X}}_{\mathrm{LE}}, \sigma^2 I\right)\right)$ for appropriate $\sigma$. If $r$ is radius of log-Euclidean geodesic ball of data, extrinsic sensitivity is given by $\Delta_{\mathrm{EX}} = 2(\exp{(r)} - 1)/n$ (Reimherr et al., 2021, Proposition 1). It should be emphasized that resultant privatized Fréchet mean is *no longer a* SPD *matrix*. Hence Reimherr et al. (2021) compared deviation between private and non private Fréchet mean in the standard Euclidean norm.

We generate random $k \times k$ SPD matrices as follows: ($i$) generate $k$ real values $(\lambda_1, \ldots, \lambda_k)$ uniformly in $[e^{-r}, e^r]$, ($ii$) build $D$ the diagonal matrix with $D_{ii} = \lambda_i$, for $i \in \{1, ..., k\}$, ($iii$) generate a $k \times k$ random orthogonal matrix $E$ with the Haar distribution, and ($iv$) build the SPD matrix as: $X = EDE^T$. This process generates SPD matrices, that can be shown to belong to the geodesic ball $\mathcal{B}_{\sqrt{k}r}(I)$ with $I$ the identity matrix: $\|\mathrm{Logm}\, X\|_F = \sqrt{\sum_{i=1}^k (\ln \lambda_i)^2} \leq \sqrt{kr^2} = \sqrt{k}r$. We use $n = 500$ and $r = 1/4$ in our experiments and hence $\Delta_{\mathrm{LE}} \leq \sqrt{k}/1000$.

Fig. 1 (first) compares utilities using a fixed $\delta = 10^{-6}$ for our mechanism, a MCMC burn-in of $50,000$ for the Riemannian Laplace mechanism, and different values of $k \in \{2, 5, 10, 15, 20, 25, 30\}$ and $\epsilon \in \{0.1, 0.2, 0.3, 0.4\}$. Each experiment is repeated 10 times, the results are averaged and the band $(\mu - 2\sigma, \mu + 2\sigma)$ is shown, where $\mu$ and $\sigma$ are the mean and standard deviation, respectively, of the associated result. The $\sigma$ is small for our mechanism, and does not appear on the plots. The tangent Gaussian mechanisms (ours) yield almost $\times 10$ utility improvement for larger $k$, for each $\epsilon$. Fig. 1 (middle) shows that, as expected, our utility is not significantly impacted by different values of $\delta \in \{10^{-7}, 10^{-8}, 10^{-9}\}$. Fig. 1 (bottom) compares utilities between Extrinsic Gaussian mechanism (analytic) and tangent Gaussian mechanism (analytic) in Euclidean distance and shows proposed mechanism is better.

## 6.2 Experiments on Real-World Datasets

We run experiments on covariance descriptors of real-world images. Covariance descriptors (Tuzel et al., 2006) have been widely used for face and person recognition (Tuzel et al., 2007; Zhang & Li, 2011; Pang et al., 2008; Križaj et al., 2013; Ma et al., 2014; Cai et al., 2010; Zeng et al., 2015; Matsukawa et al., 2016), action and gesture recognition (Cirujeda & Binefa, 2014; Hussein et al., 2013; Sharaf et al., 2015), 3D shape analysis (Tabia et al., 2014; Ma et al., 2014), medical imaging (Khan et al., 2015; Cirujeda et al., 2016); and even recently as layers in neural networks (Yu & Salzmann, 2017) - which makes them interesting data to privatize.

Let $\mathcal{I} \in \mathbb{R}^{h \times w \times c}$ be an image of height $h$, width $w$ and with $c$ channels, where $c$ is 1 for gray scale images and 3 for RGB images. Let $\phi : \mathbb{R}^{h \times w \times c} \to \mathbb{R}^{hw \times k}$ be a feature extractor of dimension $k$, i.e. $\phi(\mathcal{I})(\mathbf{x})$ is a $k$-dimensional vector at each spatial coordinate $\mathbf{x}$ in the image's domain $S$. Given a small $\eta > 0$, the *covariance descriptor* $\mathsf{R}_\eta : \mathbb{R}^{h \times w \times c} \to \mathrm{SPD}(k)$ associated with $\phi$ is defined as

$$\mathsf{R}_\eta(\mathcal{I}) = \left[ \frac{1}{|\mathcal{S}|} \sum_{\mathbf{x} \in S} (\phi(\mathcal{I})(\mathbf{x}) - \mu)(\phi(\mathcal{I})(\mathbf{x}) - \mu)^T \right] + \eta.I,$$

where $\mu = |S|^{-1} \sum_{\mathbf{x} \in \mathcal{S}} \phi(\mathcal{I})(\mathbf{x})$, and $\eta.I$ ensures $\mathsf{R}_\eta(\mathcal{I}) \in \mathrm{SPD}(k)$ with $\eta$ usually set to $10^{-6}$. Our experiments follow (Tuzel et al., 2006; Jayasumana et al., 2015) and use the covariance descriptors associated with the feature vector given as $\phi(\mathcal{I})(\mathbf{x}) = \left[ x, y, \mathcal{I}, |\mathcal{I}_x|, |\mathcal{I}_y|, |\mathcal{I}_{xx}|, |\mathcal{I}_{yy}|, \sqrt{|\mathcal{I}_x|^2 + |\mathcal{I}_y|^2}, \arctan\left(\frac{|\mathcal{I}|_x}{|\mathcal{I}|_y}\right) \right]$, where $\mathbf{x} = (x, y)$, intensities derivatives are denoted by $\mathcal{I}_x, \mathcal{I}_y, \mathcal{I}_{xx}, \mathcal{I}_{yy}$ and we added the intensity values $\mathcal{I}$ for each channel compared to (Tuzel et al., 2006; Jayasumana et al., 2015). For gray scale images, $\phi(\mathcal{I})(\mathbf{x})$ is a 9-dimensional vector that makes $\mathsf{R}_\eta(\mathcal{I})$ a $9 \times 9$ SPD matrix, while for RGB images $\phi(\mathcal{I})(\mathbf{x})$ is a 11-dimensional vector that makes $\mathsf{R}_\eta(\mathcal{I})$ a $11 \times 11$ SPD matrix. We are within the assumptions of Th. 4 since such covariance descriptors belong to geodesic balls centered at $I$, as shown by the following theorem.

**Theorem 5.** *Let $\mathsf{R}_\eta(\mathcal{I})$ be the covariance descriptor associated with the feature vector $\phi(\mathcal{I})$ above.*

1. *If $\mathcal{I}$ is a gray scale image, then $\|\mathrm{Logm}\, \mathsf{R}_\eta(\mathcal{I})\|_F \leq \sqrt{9} \max\{|\ln \eta|, |\ln(14 + \eta)|\}$.*

2. *If $\mathcal{I}$ is a RGB image, then $\|\mathrm{Logm}\, \mathsf{R}_\eta(\mathcal{I})\|_F \leq \sqrt{11} \max\{|\ln \eta|, |\ln(16 + \eta)|\}$.*

Proof is given in Appenidx A.5

### 6.2.1 Experiments on medical imaging data

We use images from 4 classes of the medical imaging datasets PATHMNIST (gray scale) and OctoMNIST (RGB) from MedMNISTv2 (Yang et al., 2021), compute the 4 class-wise Fréchet means of their covariance

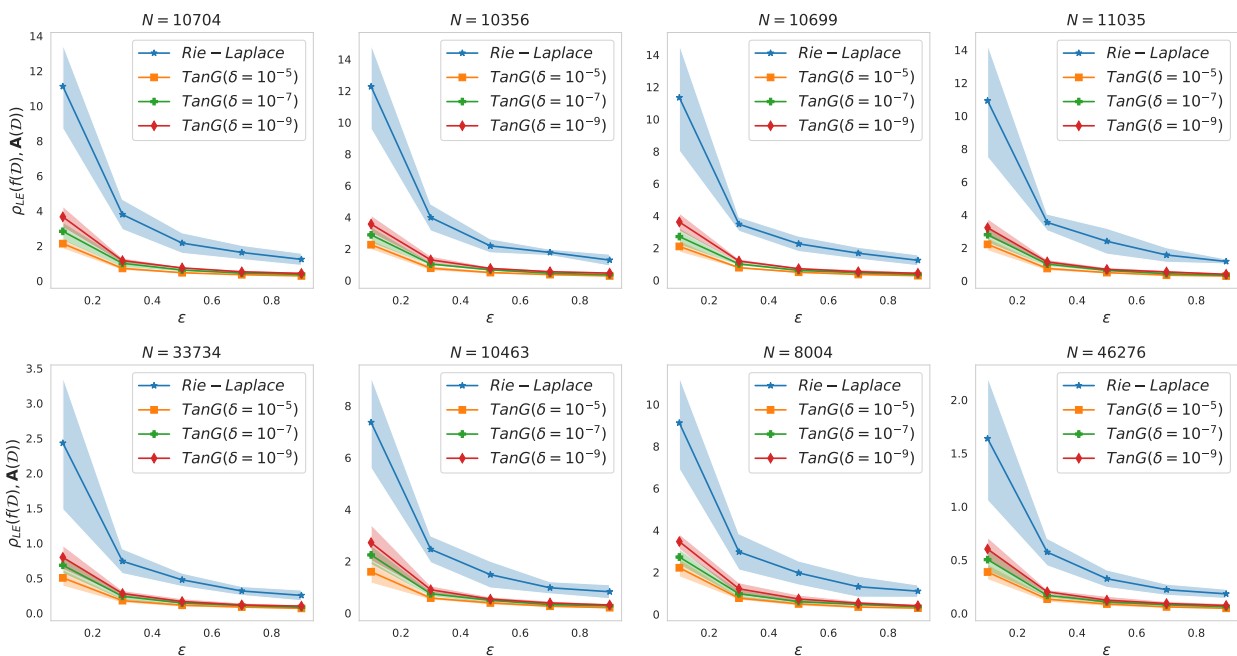

Figure 2: Utilities on the private Fréchet means for different privacy parameters $\epsilon$, and real-world datasets of sizes $N$. Top: PathMNIST (RGB images yielding $11 \times 11$ SPD descriptors). Bottom: OctoMNIST (gray scale images yielding $9 \times 9$ SPD matrices). *Rie-Laplace* is the Riemannian Laplacian mechanism (Reimherr et al., 2021) and *TanG* the tangent Gaussian mechanism for different values of $\delta$ (ours). We also show the $(\mu - 2\sigma, \mu + 2\sigma)$ band.

descriptors ($\eta = 10^{-6}$), which we privatize using the Riemannian Laplace and tangent Gaussian (analytical) mechanisms. We avoid using extrinsic approach because extrinsic sensitivity is extremely high Fig. 2 shows the utilities for different values of $\epsilon \in \{0.1, 0.3, 0.5, 0.7, 0.9\}$ and $\delta \in \{10^{-5}, 10^{-7}, 10^{-9}\}$. The datasets sizes $N$ range from 8000 to 46276 images. The sensitivity of the Fréchet mean, required for the mechanisms, is calculated using Th. 5 and Th. 3. Each experiment is repeated 10 times and averaged and the band $(\mu - 2\sigma, \mu + 2\sigma)$ is shown, where $\mu$ and $\sigma$ are the mean and standard deviation, respectively, of the associated result. Our mechanism also outperforms the Riemannian Laplace on real-world datasets, and the utility gap is higher for smaller values of $N$ and $\epsilon$.

### 6.2.2 Experiments on standard imaging data

In this section, we perform additional experiments on standard image datasets. We choose MNIST, KMNIST (Clanuwat et al., 2018) (gray scale images) and CIFAR10, FashionMNIST (Xiao et al., 2017) (RGB images) as datasets. We extract images from 4 classes for each dataset and compute the corresponding class-wise Fréchet means of their covariance descriptors ($\eta = 10^{-6}$), which we privatize using the Riemannian Laplace Mechanism (Reimherr et al., 2021) and our proposed mechanism tangent Gaussian (Analytic). Fig. 3 shows the utilities for different values of $\epsilon \in \{0.1, 0.3, 0.5, 0.7, 0.9\}$ and $\delta \in \{10^{-5}, 10^{-7}, 10^{-9}\}$. Each experiment is repeated 10 times, the results are averaged and the band $(\mu - 2\sigma, \mu + 2\sigma)$ is shown, where $\mu$ and $\sigma$ are the mean and standard deviation, respectively, of the associated result. Fig. 3 illustrates the better utility of our mechanism compared to the Riemannian Laplace mechanism.

## 7 Conclusion and Future Work

Differential privacy for geometric statistics and learning is at a very early stage. We proposed a tangent Gaussian mechanism that is specific to the SPD manifold equipped with the log-Euclidean metric, and that outperforms the only existing baseline. One limitation of our work is that the proposed mechanism is restricted to one manifold with one specific metric. While the log-Euclidean metric is one of the most

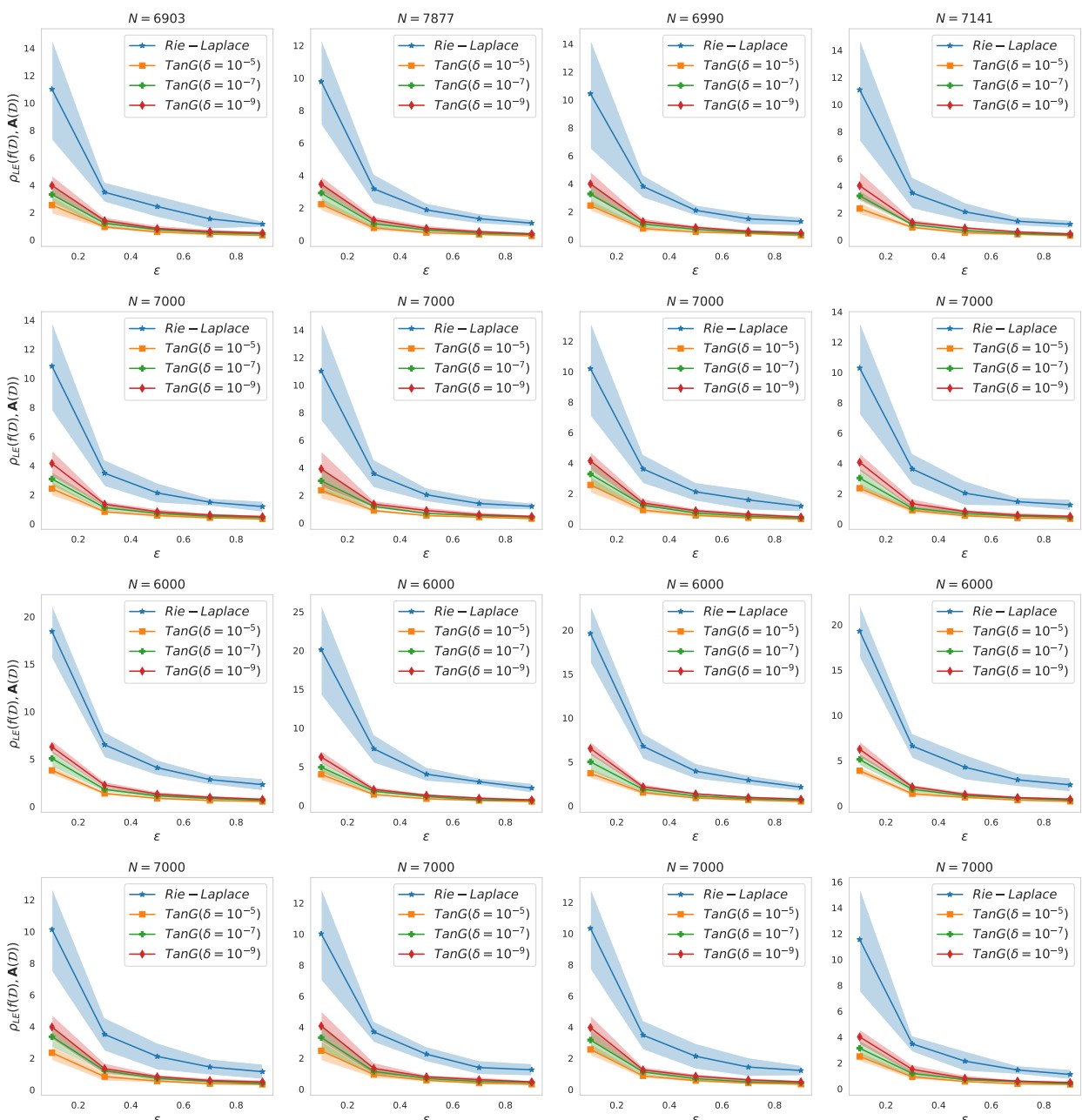

Figure 3: Utilities on the private Fréchet means for different privacy parameters $\epsilon$, and real-world datasets of sizes $N$. First and Second Row: Fréchet mean from MNIST, KMNIST (Gray scale images yielding $9 \times 9$ SPD descriptors). Third and Fourth Row: Fréchet mean from CIFAR10, FashionMNIST (RGB images yielding $11 \times 11$ SPD descriptor). *Rie-Laplace* means the Riemannian Laplacian mechanism. *TanG* means the tangent Gaussian Mechanism for different values of $\delta$ (ours). We also show the mean-2∗std, mean+2∗std bands.

important metrics on the SPD manifold, future work should investigate how to build a Gaussian mechanism that works on any complete Riemannian manifold. We could define such as a mechanism using a Riemannian Gaussian distribution derived in Pennec (2006). The main challenge would be to show that the associated procedure is $(\epsilon, \delta)$ differentially private. Future work can also seek to privatize other geometric statistical algorithms like geodesic regression or principal geodesic analysis.

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

# A   Proofs

Consider $k \in \mathbb{N}^*$. In this supplementary material, $\|.\|_{L2}$ and $\langle,\rangle_2$ denote the standard Euclidean inner product and the Euclidean norm on vectors. i.e., for all $x, y \in \mathbb{R}^k$

$$\langle x, y \rangle_{L2} = \sum_{i=1}^{p} x_i y_i.$$
$$\|x\|_{L2} = \sqrt{\langle x, x \rangle_{L2}}.$$

Then, $\langle,\rangle_F, \|.\|_F$ denotes Frobenius inner product and Frobenius norm respectively, i.e., given $A, B \in \mathbb{R}^{k \times k}$

$$\langle A, B \rangle_F = \text{Tr}[A^T B].$$
$$\|A\|_F = \sqrt{\langle A, A \rangle_F}.$$

Lastly, $\|.\|_2$ denotes the spectral norm of matrices. i.e., for all $A \in \mathbb{R}^{k \times k}$

$$\|A\|_2 = \sup_{\|x\|_{L2} \neq 0} \frac{\|Ax\|_{L2}}{\|x\|_{L2}}.$$

## A.1   Useful Lemmas

In this section, we derive the distribution of the privacy loss. Its proof requires us to first introduce the following definitions.

**Definition 3** (Diffeomorphism and Isometry). *A diffeomorphism between two manifolds $\mathcal{M}_1$ and $\mathcal{M}_2$ is an invertible smooth function whose inverse is also smooth. A diffeomorphism $\phi$ between two Riemannian manifolds $(\mathcal{M}_1, g_1)$, $(\mathcal{M}_2, g_2)$ is called an isometry if it preserves distances i.e., $\rho_{g_1}(p, q) = \rho_{g_2}(\phi(p), \phi(q))$ for all $p, q \in \mathcal{M}_1$.*

Note that Logm is a diffeomorphism from $\text{SPD}(k)$ to $\text{SYM}(k)$ and vecd is a diffeomorphism from $\text{SYM}(k)$ to $\mathbb{R}^{\frac{k(k+1)}{2}}$, making vecd Logm a diffeomorphism from $\text{SPD}(k)$ to $\mathbb{R}^{\frac{k(k+1)}{2}}$. Importantly for our derivations in the proofs of this Subsection, the operation vecd Logm preserves the distances – making it an isometry.

**Lemma A.1** (vecd Logm is an isometry)**.** *Let* $\mathrm{Logm} : \mathrm{SPD}(k) \to \mathrm{SYM}(k)$ *be the matrix logarithm and let* $\mathrm{vecd} : \mathrm{SYM}(k) \to \mathbb{R}^{\frac{k(k+1)}{2}}$ *be defined as* $\mathrm{vecd}(X) = \left[ \mathrm{diag}(X)^T, \sqrt{2}\, \mathrm{upperdiag}(X)^T \right]^T$. *Then* $\mathrm{vecd}\, \mathrm{Logm} :$ $\mathrm{SPD}(k) \to \mathbb{R}^{\frac{k(k+1)}{2}}$ *is an isometry from* $\mathrm{SPD}(k)$ *equipped with the log-Euclidean metric to standard Euclidean space* $\mathbb{R}^{\frac{k(k+1)}{2}}$ *with standard* L2 *metric, i.e.,*

$$\rho_{\mathrm{LE}}(X_1, X_2) = \rho_{\mathrm{L2}}(\mathrm{vecd}\, \mathrm{Logm}\, X_1, \mathrm{vecd}\, \mathrm{Logm}\, X_2), \tag{7}$$

*where* $X_1, X_2 \in \mathrm{SPD}(k)$. *Hence we have that*

$$\|\mathrm{Logm}\, X\|_F = \|\mathrm{vecd}\, \mathrm{Logm}\, X\|_{\mathrm{L2}}. \tag{8}$$

*Proof.* Let $X_1, X_2$ be elements of $\mathrm{SPD}(k)$. We have:

$$\rho_{\mathrm{LE}}^2(X_1, X_2)$$
$$= \|\mathrm{Logm}\, X_1 - \mathrm{Logm}\, X_2\|_F^2$$
$$= \sum_{i,j}^k (\mathrm{Logm}\, X_1 - \mathrm{Logm}\, X_2)_{ij}^2$$
$$= \sum_{i<j}^k (\mathrm{Logm}\, X_1 - \mathrm{Logm}\, X_2)_{ij}^2 + \sum_{i>j}^k (\mathrm{Logm}\, X_1 - \mathrm{Logm}\, X_2)_{ij}^2 + \sum_{i=j}^k (\mathrm{Logm}\, X_1 - \mathrm{Logm}\, X_2)_{ij}^2$$
$$= 2. \sum_{i<j}^k (\mathrm{Logm}\, X_1 - \mathrm{Logm}\, X_2)_{ij}^2 + \sum_{i=j}^k (\mathrm{Logm}\, X_1 - \mathrm{Logm}\, X_2)_{ij}^2$$
$$= \left\| \sqrt{2}\, \mathrm{upperdiag}(\mathrm{Logm}\, X_1 - \mathrm{Logm}\, X_2) \right\|_{\mathrm{L2}}^2 + \|\mathrm{diag}\, (\mathrm{Logm}\, X_1 - \mathrm{Logm}\, X_2)\|_{\mathrm{L2}}^2$$
$$= \|\mathrm{vecd}(\mathrm{Logm}\, X_1 - \mathrm{Logm}\, X_2)\|_{\mathrm{L2}}^2$$
$$= \|\mathrm{vecd}\, \mathrm{Logm}\, X_1 - \mathrm{vecd}\, \mathrm{Logm}\, X_2\|_{\mathrm{L2}}^2$$
$$= \rho_{\mathrm{L2}}^2(\mathrm{vecd}\, \mathrm{Logm}\, X_1, \mathrm{vecd}\, \mathrm{Logm}\, X_2).$$

from which we have Eq. equation 7. Eq. equation 8 follows as

$$\|\mathrm{Logm}\, X\|_F = \rho_{\mathrm{LE}}(X, I) = \rho_{\mathrm{L2}}(\mathrm{vecd}\, \mathrm{Logm}\, X, \mathrm{vecd}\, \mathrm{Logm}\, I) = \|\mathrm{vecd}\, \mathrm{Logm}\, X\|_{\mathrm{L2}}.$$

$\square$

Now, we prove some useful properties of the Log Gaussian distribution, denoted $\mathcal{LN}$, that we will use later. Essentially, we show that the Log Gaussian distribution behaves "nicely" with vector space structure of $\mathrm{SPD}(k)$. We recall that the vector space operations on the SPD manifold are defined as follows,

$$X_1 \oplus X_2 = \mathrm{Expm}\left[\mathrm{Logm}\, X_1 + \mathrm{Logm}\, X_2\right]. \tag{9}$$
$$X_1 \ominus X_2 = \mathrm{Expm}\left[\mathrm{Logm}\, X_1 - \mathrm{Logm}\, X_2\right]. \tag{10}$$

**Lemma A.2.** *Take* $k \in \mathbb{N}$. *Let* $I$ *denote the* $k \times k$ *identity matrix, and consider* $M, C \in \mathrm{SPD}(k)$, $\Sigma \in$ $\mathrm{SPD}(\frac{k(k+1)}{2})$ *and* $\chi_d^2$ *the chi-square distribution with* $d$ *degrees of freedom. Then:*

$$X \sim \mathcal{LN}(I, \Sigma) \implies X \oplus M \sim \mathcal{LN}(M, \Sigma). \tag{11}$$
$$X \sim \mathcal{LN}(I, \sigma^2 I) \implies \langle \mathrm{Logm}\, C, \mathrm{Logm}\, X \rangle_F \sim \mathcal{N}(0, \sigma^2 \|\mathrm{Logm}\, C\|_F^2). \tag{12}$$
$$X \sim \mathcal{LN}(I, \sigma^2 I) \implies \|\mathrm{Logm}\, X\|_F^2 \sim \sigma^2 \chi_{\frac{k(k+1)}{2}}^2. \tag{13}$$

*Proof.* We first recall standard properties of multivariate normal distribution. Let $m, a \in \mathbb{R}^p$ and $\Sigma, I \in \mathbb{R}^{p \times p}$ then following properties hold true.

$$x \sim \mathcal{N}(m, \Sigma) \implies a + x \sim \mathcal{N}(a + m, \Sigma). \tag{14}$$

$$x \sim \mathcal{N}(m, \Sigma) \implies a^T x \sim \mathcal{N}(a^T m, a^T \Sigma a). \tag{15}$$

$$x \sim \mathcal{N}(0, \sigma^2 I) \implies \|x\|_{\mathrm{L2}}^2 \sim \sigma^2 \chi_p^2. \tag{16}$$

where $\chi^2$ denotes chi-square distribution. We prove the properties (a)-(c) below.

(a) Distribution of $X \oplus M$.

$$
\begin{aligned}
\mathrm{vecd}[\mathrm{Logm}[X \oplus M]] &\overset{(*)}{=} \mathrm{vecd}[\mathrm{Logm}[\mathrm{Expm}\left[\log X + \log M\right]]] \\
&= \mathrm{vecd}[\mathrm{Logm}\, X + \mathrm{Logm}\, M] \\
&= \mathrm{vecd}[\mathrm{Logm}\, X] + \mathrm{vecd}[\mathrm{Logm}\, M] \\
&\overset{(**)}{\sim} \mathcal{N}(\mathrm{vecd}[\mathrm{Logm}\, M], \Sigma).
\end{aligned}
$$

where in $(*)$ we used Eq. 9 and in $(**)$ Eq. equation 14.

(b) Distribution of $\langle \mathrm{Logm}\, C, \mathrm{Logm}\, X \rangle_F$.

$$
\begin{aligned}
\langle \mathrm{Logm}\, C, \mathrm{Logm}\, X \rangle_F &\overset{(*)}{=} \langle \mathrm{vecd}[\mathrm{Logm}\, C], \mathrm{vecd}[\mathrm{Logm}\, X] \rangle_{\mathrm{L2}} \\
&\overset{(**)}{\sim} \mathcal{N}\left(\langle \mathrm{vecd}[\mathrm{Logm}\, C], 0 \rangle_{\mathrm{L2}}, \mathrm{vecd}[\mathrm{Logm}\, C]^T \sigma^2 I \, \mathrm{vecd}[\mathrm{Logm}\, C]\right) \\
&\sim \mathcal{N}(0, \sigma^2 \|\mathrm{vecd}[\mathrm{Logm}\, C]\|_{\mathrm{L2}}^2) \\
&\overset{(*)}{\sim} \mathcal{N}(0, \sigma^2 \|\mathrm{Logm}\, C\|_F^2).
\end{aligned}
$$

where we used Eq. 8 in $(*)$ and Eq. 15 in $(**)$.

(c) Distribution of $\|\mathrm{Logm}\, X\|_F^2$.

$$\|\mathrm{Logm}\, X\|_F^2 \overset{(*)}{=} \|\mathrm{vecd}[\mathrm{Logm}\, X]\|_{\mathrm{L2}}^2 \overset{(**)}{\sim} \sigma^2 \chi_{\frac{k(k+1)}{2}}^2.$$

where we used Eq. 8 in $(*)$ and Eq. 16 in $(**)$ with $p = \frac{k(k+1)}{2}$. $\qquad \square$

As corollary, we give equivalent reformulation of tangent Gaussian mechanism that will useful in the rest of the proofs.

**Corollary A.3** (Equivalent Reformulation of tangent Gaussian)**.** *Let* $\mathbf{A}_{\mathrm{TG}}$ *be a tangent Gaussian mechanism defined as* $\mathbf{A}_{\mathrm{TG}}(f(\mathcal{D})) = X$, $X \sim \mathcal{LN}(f(\mathcal{D}), \sigma^2 I)$. *Then, it is equivalently defined as:*

$$\mathbf{A}_{\mathrm{TG}}(f(\mathcal{D})) = f(\mathcal{D}) \oplus N, N \sim \mathcal{LN}(I, \sigma^2 I).$$

*Proof.* The proof comes from Eq. 11 of Lemma A.2. $\qquad \square$

Now, we are ready to prove the distribution of the privacy loss of the tangent Gaussian Mechanism which is given Th. 1.

## A.2 Proof of Th. 1

**Theorem A.4** (Distribution of the privacy loss of the tangent Gaussian). *Let $\mathbf{A}_{\mathrm{TG}}$ be a tangent Gaussian mechanism with variance $\sigma^2$. Its privacy loss is normally distributed as*

$$L_{\mathbf{A}_{\mathrm{TG}},\mathcal{D},\mathcal{D}'} \sim \mathcal{N}\left(\frac{\rho^2_{\mathrm{LE}}(f(\mathcal{D}),f(\mathcal{D}'))}{2\sigma^2}, \frac{\rho^2_{\mathrm{LE}}(f(\mathcal{D}),f(\mathcal{D}'))}{\sigma^2}\right).$$

*Proof.* Assume that $\mathcal{D},\mathcal{D}'$ are adjacent datasets. Let $V = f(\mathcal{D}) \ominus f(\mathcal{D}')$. Consider the privacy loss random variable $L_{\mathbf{A}_{\mathrm{TG}},\mathcal{D},\mathcal{D}'}$. Let $Y = \mathbf{A}_{\mathrm{TG}}(\mathcal{D})$.

$$\ln\left(\frac{p_{\mathbf{A}_{\mathrm{TG}}(\mathcal{D})}(Y)}{p_{\mathbf{A}_{\mathrm{TG}}(\mathcal{D}')}(Y)}\right)$$

$$\overset{(1)}{=} \ln\left(\frac{p_{\mathbf{A}_{\mathrm{TG}}(\mathcal{D})}(f(\mathcal{D}) \oplus N)}{p_{\mathbf{A}_{\mathrm{TG}}(\mathcal{D}')}(f(\mathcal{D}) \oplus N)}\right)$$

$$\overset{(2)}{=} -\frac{1}{2}\left[\mathrm{vecd}\left(\mathrm{Logm}(f(\mathcal{D}) \oplus N) - \mathrm{Logm}\, f(D)\right)\right]^T \frac{I}{\sigma^2} \mathrm{vecd}\left(\mathrm{Logm}(f(\mathcal{D}) \oplus N) - \mathrm{Logm}\, f(D)\right)$$
$$+ \frac{1}{2}\left[\mathrm{vecd}\left(\mathrm{Logm}(f(\mathcal{D}) \oplus N) - \mathrm{Logm}\, f(D')\right)\right]^T \frac{I}{\sigma^2} \mathrm{vecd}\left(\mathrm{Logm}(f(\mathcal{D}) \oplus N) - \mathrm{Logm}\, f(D')\right)$$

$$\overset{(3)}{=} -\frac{1}{2\sigma^2}\left\|\mathrm{vecd}\left(\mathrm{Logm}\, N\right)\right\|^2_{\mathrm{L2}} + \frac{1}{2\sigma^2}\left\|\mathrm{vecd}\left(\mathrm{Logm}\, f(D) - \mathrm{Logm}\, f(D') + \mathrm{Logm}\, N\right)\right\|^2_{\mathrm{L2}}$$

$$\overset{(4)}{=} -\frac{1}{2\sigma^2}\left\|\mathrm{vecd}\left(\mathrm{Logm}\, N\right)\right\|^2_{\mathrm{L2}} + \frac{1}{2\sigma^2}\left\|\mathrm{vecd}\left(\mathrm{Logm}(V \oplus N)\right)\right\|^2_{\mathrm{L2}}$$

$$\overset{(5)}{=} \frac{1}{2\sigma^2}\left[\|\mathrm{Logm}(V \oplus N)\|^2_F - \|\mathrm{Logm}\, N\|^2_F\right]$$

$$= \frac{1}{2\sigma^2}\left[\|\mathrm{Logm}\, V\|^2_F + 2\langle\mathrm{Logm}\, V, \mathrm{Logm}\, N\rangle_F\right]$$

$$\overset{(6)}{\sim} \frac{1}{2\sigma^2}\left[\|\mathrm{Logm}\, V\|^2_F + 2\mathcal{N}\left(0, \sigma^2\|\mathrm{Logm}\, V\|^2_F\right)\right]$$

$$\overset{(7)}{\sim} \mathcal{N}\left(\frac{\|\mathrm{Logm}\, V\|^2_F}{2\sigma^2}, \frac{\|\mathrm{Logm}\, V\|^2_F}{\sigma^2}\right)$$

$$\overset{(8)}{\sim} \mathcal{N}\left(\frac{\rho^2_{\mathrm{LE}}(f(\mathcal{D}),f(\mathcal{D}'))}{2\sigma^2}, \frac{\rho^2_{\mathrm{LE}}(f(\mathcal{D}),f(\mathcal{D}'))}{\sigma^2}\right),$$

where we used following properties in each of the steps labeled above.

1. Equivalent reformulation of tangent Gaussian, Corollary. A.3.

2. Density of Log Gaussian Distribution.

3. $f(\mathcal{D}) \oplus N = \mathrm{Expm}[\mathrm{Logm}\, f(\mathcal{D}) + \mathrm{Logm}\, N]$.

4. $\mathrm{Logm}(V \oplus N) = \mathrm{Logm}\, f(D) - \mathrm{Logm}\, f(D') + \mathrm{Logm}\, N$.

5. Isometry of the vecd operation, Eq.8

6. Eq. 12 in Lemma. A.2.

7. standard Gaussian property, see Eq. 14.

8. $\|\mathrm{Logm}\, V\|^2_F = \|\mathrm{Logm}\, f(\mathcal{D}) - \mathrm{Logm}\, f(\mathcal{D}')\|^2_F = \rho^2_{\mathrm{LE}}(f(\mathcal{D}),f(\mathcal{D}'))$.

$\square$

### A.3 Proof of Th. 2

In this section we give proof of privacy guarantee of the tangent Gaussian Mechanism.

*Proof.* The proof proceeds similarly to the proofs referenced below, by only replacing the standard sensitivity $\Delta_{\text{L2}}$ with respect to the Euclidean $L_2$ metric, by $\Delta_{\text{LE}}$:

1. (Classical). See Th. A.1 (Appendix A Page 261) in Dwork et al. (2014).

2. (Analytic). See Th. 5, Th. 8, Th. 9 (Section 3) in Balle & Wang (2018).

The fact that mechanism is manifold valued comes into play while deriving privacy loss (Taken care by Theorem 1). Once privacy loss (which is *real valued scalar* random variable) is derived, going from privacy loss to actual privacy guarantee wouldn't be affected whether a mechanism is manifold-valued or not because both of the above proofs *entirely* rely on properties of one-dimensional euclidean Gaussian random variables. Specifically,

1. (Classical). Directly employs tail bound of one-dimensional Gaussian variable that $\mathbb{P}[x > t] < \frac{\sigma}{\pi} \exp(-\frac{t^2}{2\sigma^2})$

2. (Analytic). The method employs both the sufficient and necessary conditions of the $(\epsilon, \delta)$ guarantee. Additionally, the algorithm avoids using tail bounds, since they may be loose, instead uses properties of Gaussian CDFs and employs binary search to solve analytically for $\sigma$, given $(\epsilon, \delta)$. See (Balle & Wang, 2018, Algorithm 1) and discussion therein for more details.

$\square$

### A.4 Proof of Th. 3 and Th. 4

In this section, we prove the sensitivity of the Fréchet Mean in Theorem. 3 and then the utility of the tangent Gaussian Mechanism in Theorem. 4. First we give the proof of 3.

*Proof.* Consider $k \in \mathbb{N}$, $0 < r < \infty$ and $M \in \text{SPD}(k)$ such that $\mathcal{B}_r(M)$ is a geodesic ball of radius $r$ and center $M$. Let $\mathcal{D} \sim \mathcal{D}'$ be adjacent datasets of size $n \in \mathbb{N}$ that lie in $\mathcal{B}_r(M)$. Without loss of generality, we can assume that they differ only by their last data point $X_n$ and $X_n'$: $\mathcal{D} = \{X_1, X_2, \ldots, X_n\}$ and $\mathcal{D}' = \{X_1, X_2, \ldots, X_n'\}$. Let $\overline{X}_{\mathcal{D}}, \overline{X}_{\mathcal{D}'}$ denote the Fréchet means of $\mathcal{D}$ and $\mathcal{D}'$ for the log-Euclidean metric, which can be expressed in closed forms as mentioned in the main text. The log-Euclidean distance between the Fréchet means writes:

$$
\begin{aligned}
&\rho_{\text{LE}}(\overline{X}_{\mathcal{D}}, \overline{X}_{\mathcal{D}'}) \\
&\stackrel{(*)}{=} \left\| \text{Logm}\left(\text{Expm}\left(\sum_{i=1}^{n} \frac{\text{Logm}\, X_i}{n}\right)\right) - \text{Logm}\left(\text{Expm}\left(\sum_{i=1}^{n-1} \frac{\text{Logm}\, X_i}{n} + \frac{\text{Logm}\, X_n'}{n}\right)\right) \right\|_F \\
&= \left\| \frac{1}{n}\sum_{i=1}^{n-1} \text{Logm}\, X_i - \frac{1}{n}\sum_{i=1}^{n-1} \text{Logm}\, X_i + \frac{1}{n}\text{Logm}\, X_n - \frac{1}{n}\text{Logm}\, X_n' \right\|_F \\
&= \frac{1}{n}\left\| \text{Logm}\, X_n - \text{Logm}\, X_n' \right\|_F \\
&= \frac{1}{n}\rho_{\text{LE}}(X_n, X_n').
\end{aligned}
$$

$$\Delta_{\text{LE}} = \sup_{\mathcal{D} \sim \mathcal{D}'} \rho_{\text{LE}}(\overline{X}_{\mathcal{D}}, \overline{X}_{\mathcal{D}'}) = \sup_{\mathcal{D} \sim \mathcal{D}'} \frac{1}{n} \rho_{\text{LE}}(X_n, X_n') \overset{(\dagger)}{\leq} \frac{1}{n} \left[ \rho_{\text{LE}}(X_n, M) + \rho_{\text{LE}}(M, X_n') \right] \overset{(\ddagger)}{\leq} \frac{2r}{n},$$

where we use the closed form for the log-Euclidean Fréchet means in $(*)$, the triangle inequality in $(\dagger)$ and assumption that data lies in $\mathcal{B}_r(M)$ in $(\ddagger)$. $\qquad\square$

Proof of Th. 4 is given as follows,

*Proof.* Consider deviation $\rho_{\text{LE}}^2(f(\mathcal{D}), \mathbf{A}_{\text{TG}}(\mathcal{D})))$

$$\rho_{\text{LE}}^2(f(\mathcal{D}), \mathbf{A}_{\text{TG}}(\mathcal{D}))) = \|\text{Logm}\, f(\mathcal{D}) - \text{Logm}\, \mathbf{A}_{\text{TG}}(\mathcal{D})\|_F^2 \overset{(1)}{=} \|\text{Logm}\, f(\mathcal{D}) - \text{Logm}(f(\mathcal{D}) \oplus N)\|_F^2$$
$$\overset{(2)}{=} \|\text{Logm}\, N\|_F^2$$
$$\overset{(3)}{\sim} \sigma^2 \chi_d^2,$$

where we use the following properties at each step:

(1) Corollary. A.3.

(2) $f(\mathcal{D}) \oplus N = \text{Expm}\left[\text{Logm}\, f(\mathcal{D}) + \text{Logm}\, N\right]$.

(3) Eq. 13 of Lemma. A.2.

Now we derive expression for $\mathbb{E}[\rho_{\text{LE}}^2(f(\mathcal{D}), \mathbf{A}_{\text{TG}}(\mathcal{D}))]$

$$\mathbb{E}[\rho_{\text{LE}}^2(f(\mathcal{D}), \mathbf{A}_{\text{TG}}(\mathcal{D}))] \overset{(1)}{=} \sigma^2 d$$
$$\overset{(2)}{=} \frac{2\Delta_{\text{LE}}^2 \ln(1.25/\delta)d}{\epsilon^2}$$
$$\overset{(3)}{\leq} \frac{8r^2 \ln(1.25/\delta)d}{n^2 \epsilon^2}.$$

where we use following properties at each step:

1. $c \sim \chi_d^2 \implies \mathbb{E}[c] = d$ i.e., expectation of chi squared distributed random variable is number of degrees of freedom.

2. $\sigma = \Delta_{\text{LE}} \sqrt{2 \ln(1.25/\delta)}/\epsilon$ for $(\epsilon, \delta)$-$\mathbf{A}_{\text{TG}}$ from Th. 2.

3. $\Delta_{\text{LE}} \leq \frac{2r}{n}$ from Th. 3.

$\qquad\square$

## A.5 Proof of Theorem 5

In this section we derive log-Euclidean geodesic radius of covariance descriptors. We first prove following lemma that relates $\|\text{Logm}\, X\|_F$ in terms of lower bound on least eigenvalue and upper bound on largest eigenvalue of $X$.

**Lemma A.5.** *If $X \in \text{SPD}(k)$ and let $\lambda_{\min}(X), \lambda_{\max}(X)$ be the minimum and maximum eigenvalues of $X$. If $\ell \leq \lambda_{\min}(X)$ and $\lambda_{\max}(X) \leq L$ Then, $\|\text{Logm}\, X\|_F \leq \sqrt{k} \max\{|\ln \ell|, |\ln L|\}$.*

*Proof.* Consider,

$$
\begin{aligned}
\|\text{Logm}\, X\|_F &\overset{(\dagger)}{\leq} \sqrt{k}\, \|\text{Logm}\, X\|_2 \\
&= \sqrt{k}\, \max_{i=1}^{n} |\ln \lambda_i| \\
&= \sqrt{k}\, \max\left\{ |\min_{i=1}^{n} \ln \lambda_i|, |\max_{i=1}^{n} \ln \lambda_i| \right\} \\
&\overset{(\ddagger)}{=} \sqrt{k}\, \max\left\{ |\ln \min_{i=1}^{n} \lambda_i|, |\ln \max_{i=1}^{n} \lambda_i| \right\} \\
&= \sqrt{k}\, \max\left\{ |\ln \lambda_{\min}|, |\ln \lambda_{\max}| \right\}.
\end{aligned}
\tag{17}
$$

where $(\dagger)$ uses the fact that $A \in \mathbb{R}^{k \times k}, \|A\|_F \leq \sqrt{k}\, \|A\|_2$ and $(\ddagger)$ uses the fact that $\ln$ is monotonically increasing. Now, we split the derivation into two cases.

1. CASE $\lambda_{\min}(X) \geq 1$. For $x \geq 1$, $|\ln x|$ is an increasing function, which gives us: $|\ln \ell| \leq |\ln \lambda_{\min}(X)| \leq |\ln \lambda_{\max}| \leq |\ln L|$

$$
\sqrt{k}\, \max\left\{ |\ln \lambda_{\min}|, |\ln \lambda_{\max}| \right\} \leq \sqrt{k}|\ln L| = \sqrt{k}\, \max\left\{ |\ln \ell|, |\ln L| \right\}.
\tag{18}
$$

2. CASE $\lambda_{\min}(X) < 1$. For $x < 1$, $|\ln x|$ is a decreasing function: $|\ln \lambda_{\min}| \leq |\ln \ell|$. We further split the derivation into two sub-cases here

   (a) SUB-CASE $\lambda_{\max} \geq 1$. In this sub-case $|\ln \lambda_{\max}| \leq |\ln L|$ and $\ln \lambda_{\min} \leq |\ln \ell|$ from which we have that

   $$
   \sqrt{k}\, \max\left\{ |\ln \lambda_{\min}|, |\ln \lambda_{\max}| \right\} \leq \sqrt{k}\, \max\left\{ |\ln \ell|, |\ln L| \right\}.
   \tag{19}
   $$

   (b) SUB-CASE $\lambda_{\max} < 1$. In this sub-case $|\ln L| \leq |\ln \lambda_{\max}| \leq |\ln \lambda_{\min}| \leq |\ln \ell|$.

   $$
   \sqrt{k}\, \max\left\{ |\ln \lambda_{\min}|, |\ln \lambda_{\max}| \right\} \leq \sqrt{k}|\ln \ell| = \sqrt{k}\, \max\left\{ |\ln \ell|, |\ln L| \right\}.
   \tag{20}
   $$

   Based on Eq. 18, Eq. 19, Eq. 20 and Eq.17. We can conclude the lemma.

$\square$

**Lemma A.6.** *Let $R_\eta(\mathcal{I})$ denote the covariance descriptor for image $\mathcal{I}$ for given $\eta > 0$, which is defined as follows ,*

$$
R_\eta(\mathcal{I}) = \left[ \frac{1}{|\mathcal{S}|} \sum_{\boldsymbol{x} \in S} (\phi(\mathcal{I})(\boldsymbol{x}) - \mu)(\phi(\mathcal{I})(\boldsymbol{x}) - \mu)^T \right] + \eta.I,
$$

*with,*

$$
\phi(\mathcal{I}) = \left[ x, y, \mathcal{I}, |\mathcal{I}_x|, |\mathcal{I}_y|, |\mathcal{I}_{xx}|, |\mathcal{I}_{yy}|, \sqrt{|\mathcal{I}_x|^2 + |\mathcal{I}_y|^2}, \arctan\left( \frac{|\mathcal{I}_x|}{|\mathcal{I}_y|} \right) \right].
$$

*where $x, y$ are grid positions of Image $\mathcal{I}$, $\mathcal{I}$ denote pixel intensity values , $|\mathcal{I}_x|, |\mathcal{I}_y|$ denotes first order intensity derivatives and $|\mathcal{I}_{xx}|, |\mathcal{I}_{yy}|$ denotes the second order intensity derivatives then following holds,*

1. *If $\mathcal{I}$ is grayscale image, then $\|R_\eta(\mathcal{I})\|_2 \leq 12 + \eta$.*

2. *If $\mathcal{I}$ is RGB image then $\|R_\eta(\mathcal{I})\|_2 \leq 14 + \eta$.*

*Proof.* We have:

$$
\begin{aligned}
\|\mathsf{R}_\eta(\mathcal{I})\|_2 &= \left\| \left[ \frac{1}{|\mathcal{S}|} \sum_{\mathbf{x} \in S} (\phi(\mathcal{I})(\mathbf{x}) - \mu)(\phi(\mathcal{I})(\mathbf{x}) - \mu)^T \right] + \eta.I \right\|_2 \\
&\overset{(1)}{\leq} \frac{1}{|\mathcal{S}|} \sum_{\mathbf{x} \in S} \left\| (\phi(\mathcal{I})(\mathbf{x}) - \mu)(\phi(\mathcal{I})(\mathbf{x}) - \mu)^T \right\|_2 + \|\eta.I\|_2 \\
&\leq \max_{\mathbf{x} \in \mathcal{S}} \left\| (\phi(\mathcal{I})(\mathbf{x}) - \mu)(\phi(\mathcal{I})(\mathbf{x}) - \mu)^T \right\|_2 + \eta \\
&\overset{(2)}{=} \max_{\mathbf{x} \in \mathcal{S}} \left\| (\phi(\mathcal{I})(\mathbf{x}) - \mu) \right\|_{\mathrm{L2}}^2 + \eta \\
&\overset{(3)}{\leq} \max_{\mathbf{x} \in \mathcal{S}} \| \phi(\mathcal{I})(\mathbf{x}) \|_{\mathrm{L2}}^2 + \eta,
\end{aligned}
\tag{21}
$$

where we used following properties in each of the steps:

1. Triangle Inequality.

2. For all $a \in \mathbb{R}^p$, the spectral norm of 1-rank matrix $aa^T$ is $\|a\|_{\mathrm{L2}}^2$.

3. Consider the descriptor $\phi(\mathcal{I}) = \left[ x, y, \mathcal{I}, |\mathcal{I}_x|, |\mathcal{I}_y|, |\mathcal{I}_{xx}|, |\mathcal{I}_{yy}|, \sqrt{|\mathcal{I}_x|^2 + |\mathcal{I}_y|^2}, \arctan\left(\frac{|\mathcal{I}_x|}{|\mathcal{I}_y|}\right) \right]$. Then, $\phi(\mathcal{I})(\mathbf{x})_i \geq 0$ for each $\mathbf{x} \in \mathcal{S}$ and $i \in \{1, \ldots, k\}$. This yields: $(\mu)_i = \left( |\mathcal{S}|^{-1} \sum_{\mathbf{x} \in \mathcal{S}} \phi(\mathcal{I})(\mathbf{x}) \right)_i \geq 0$. Hence it implies that $\|\phi(\mathcal{I})(\mathbf{x}) - \mu\|_{\mathrm{L2}}^2 \leq \|\phi(\mathcal{I})(\mathbf{x})\|_{\mathrm{L2}}^2$.

Then, the following calculations provide an upper bound for $\|\phi(\mathcal{I})(\mathbf{x})\|_2^2$. Specifically, we bound each of the 6 elements constituting the descriptor $\phi(\mathcal{I}) = \left[ x, y, \mathcal{I}, |\mathcal{I}_x|, |\mathcal{I}_y|, |\mathcal{I}_{xx}|, |\mathcal{I}_{yy}|, \sqrt{|\mathcal{I}_x|^2 + |\mathcal{I}_y|^2}, \arctan\left(\frac{|\mathcal{I}_x|}{|\mathcal{I}_y|}\right) \right]$.

1. Normalized grid positions : $\forall \mathbf{x} \in \mathcal{S}$, $0 \leq x, y \leq 1$.

2. Pixel intensity values $C_i$ for $i \in [c]$ : $\forall \mathbf{x} \in \mathcal{S}$, $0 \leq C_i[\mathbf{x}] \leq 1$.

3. First intensity derivatives $|\mathcal{I}_x|, |\mathcal{I}_y|$: The first intensity derivatives can be obtained by the convolution operation (denoted as $\star$):

$$
\mathcal{I}_x = \mathcal{I} \star \frac{1}{4} \begin{bmatrix} +1 & 0 & -1 \\ +2 & 0 & -2 \\ +1 & 0 & -1 \end{bmatrix}, \mathcal{I}_y = \mathcal{I} \star \frac{1}{4} \begin{bmatrix} +1 & +2 & +1 \\ 0 & 0 & 0 \\ -1 & -2 & -1 \end{bmatrix}.
$$

Since $0 \leq \mathcal{I}(\mathbf{x}) \leq 1$, using the definition of the convolution operation yields $\forall \mathbf{x} \in \mathcal{S}$, $|\mathcal{I}_x(\mathbf{x})| \leq 1$, $|\mathcal{I}_y(\mathbf{x})| \leq 1$.

4. Second intensity derivatives $|\mathcal{I}_{xx}|, |\mathcal{I}_{yy}|$ : The second intensity derivatives can be obtained by the convolution operation (denoted as $\star$)

$$
\mathcal{I}_{xx} = \mathcal{I} \star \frac{1}{32} \begin{bmatrix} +1 & 0 & -2 & 0 & 1 \\ +4 & 0 & -8 & 0 & 4 \\ +6 & 0 & -12 & 0 & 6 \\ +4 & 0 & -8 & 0 & 4 \\ +1 & 0 & -2 & 0 & 1 \end{bmatrix}, \mathcal{I}_{yy} = \mathcal{I} \star \frac{1}{32} \begin{bmatrix} +1 & +4 & +6 & +4 & +1 \\ 0 & 0 & 0 & 0 & 0 \\ -2 & -8 & -12 & -8 & -2 \\ 0 & 0 & 0 & 0 & 0 \\ +1 & +4 & +6 & +4 & +1 \end{bmatrix}.
$$

Since $0 \leq \mathcal{I}(\mathbf{x}) \leq 1$, using the definition of the convolution operation yields $|\mathcal{I}_{xx}(\mathbf{x})| \leq 1$, $|\mathcal{I}_{yy}(\mathbf{x})| \leq 1$.

5. Norm of first intensity derivatives : since $|\mathcal{I}_x(\mathbf{x})| \leq 1$, $|\mathcal{I}_y(\mathbf{x})| \leq 1$ we have that $\forall \mathbf{x} \in \mathcal{S}, \sqrt{\mathcal{I}_x(\mathbf{x})^2 + \mathcal{I}_y(\mathbf{x})^2} \leq \sqrt{2}$.

6. Angle of intensity derivatives : Note that for $a \geq 0$, $0 \leq \arctan a \leq \frac{\pi}{2}$. Hence we have that $\forall \mathbf{x} \in \mathcal{S}, \arctan\left(\left|\frac{\mathcal{I}_x(\mathbf{x})}{\mathcal{I}_y(\mathbf{x})}\right|\right) \leq \frac{\pi}{2}$.

These provide the following upper bounds on L2 norm of $\phi(\mathcal{I})(\mathbf{x})$,

$$\text{for a gray scale image}, \forall \mathbf{x} \in \mathcal{S} \, \|\phi(\mathcal{I})(\mathbf{x})\|_{\text{L2}}^2 \leq 12, \tag{22}$$

$$\text{for RGB image}, \forall \mathbf{x} \in \mathcal{S} \, \|\phi(\mathcal{I})(\mathbf{x})\|_{\text{L2}}^2 \leq 14. \tag{23}$$

The claim follows by using Eq. 22, Eq. 23 in Eq. 21 $\qquad \square$

**Theorem A.7** (Geodesic Radius of Covariance Descriptors)**.**

1. *If $\mathcal{I}$ is a gray scale image, then $\|\text{Logm}\, R_\eta(\mathcal{I})\|_F \leq \sqrt{9} \max\{|\ln \eta|, |\ln(12 + \eta)|\}$.*

2. *If $\mathcal{I}$ is a RGB image, then $\|\text{Logm}\, R_\eta(\mathcal{I})\|_F \leq \sqrt{11} \max\{|\ln \eta|, |\ln(14 + \eta)|\}$.*

*Proof.* We first note that

$$
\begin{aligned}
\lambda_{\min}(R_\eta(\mathcal{I})) &= \lambda_{\min}\left[\frac{1}{|\mathcal{S}|}\sum_{\mathbf{x} \in S}(\phi(\mathcal{I})(\mathbf{x}) - \mu)(\phi(\mathcal{I})(\mathbf{x}) - \mu)^T + \eta.I\right] \\
&\overset{(1)}{\geq} \lambda_{\min}\left[\frac{1}{|\mathcal{S}|}\sum_{\mathbf{x} \in S}(\phi(\mathcal{I})(\mathbf{x}) - \mu)(\phi(\mathcal{I})(\mathbf{x}) - \mu)^T\right] + \lambda_{\min}[\eta I] \\
&\overset{(2)}{\geq} 0 + \eta.
\end{aligned}
\tag{24}
$$

where we used Weyl's inequality for symmetric matrices in (1) and $\lambda_{\min}$ of positive semi definite matrix is $\geq 0$ and $\lambda_{\min}[\eta I] = \eta$ in (2).

For gray scale images, $R_\eta(\mathcal{I})$ produces a $9 \times 9$ matrix:

$$
\begin{aligned}
\|\text{Logm}\, R_\eta(\mathcal{I})\|_F &\overset{(*)}{\leq} \sqrt{9} \max\{|\ln \ell|, |\ln L|\} \\
&\overset{(**)}{=} \sqrt{9} \max\{|\ln \eta|, |\ln(12 + \eta)|\},
\end{aligned}
$$

where we use Lemma. A.5 in $(*)$ and Eq.24$(\ell = \eta)$ and Lemma. A.6$(L = 12 + \eta)$ in $(**)$

For RGB images, $R_\eta(\mathcal{I})$ produces a $11 \times 11$ matrix:

$$
\begin{aligned}
\|\text{Logm}\, R_\eta(\mathcal{I})\|_F &\overset{(\dagger)}{\leq} \sqrt{11} \max\{|\ln \ell|, |\ln L|\} \\
&\overset{(\ddagger)}{=} \sqrt{11} \max\{|\ln \eta|, |\ln(14 + \eta)|\},
\end{aligned}
$$

where we use Lemma. A.5 in $(\dagger)$ and Eq.24 $(\ell = \eta)$ and Lemma. A.6$(L = 14 + \eta)$ in $(\ddagger)$, in a similar fashion. $\qquad \square$

Note that in all of our experiments, we choose $\eta = 10^{-6}$ and hence $|\ln \eta| \approx 13.8$ domninates over $|\ln(12 + \eta)| \approx 2.5$ and $|\ln(14 + \eta)| \approx 2.6$.

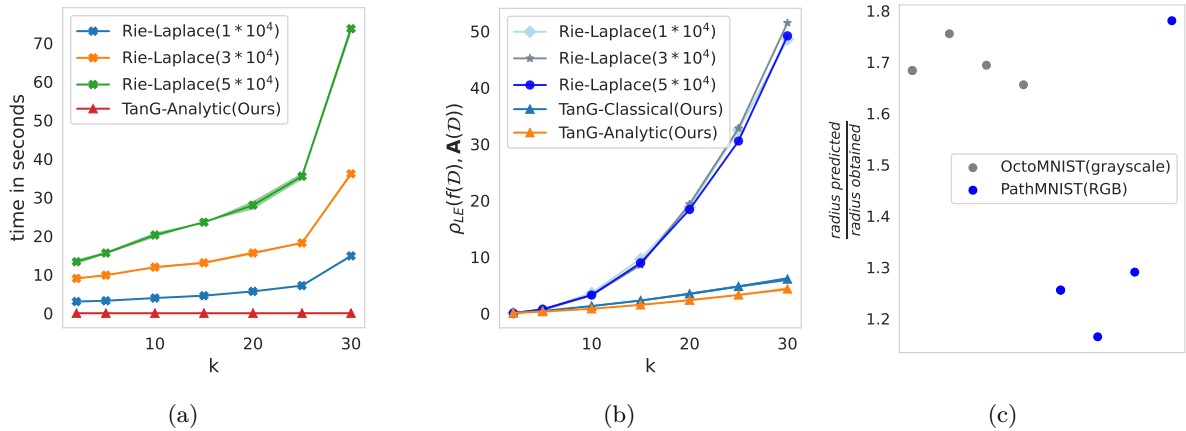

Figure 4: (a) Computational times for *Rie-Laplace*$(x)$ the Riemannian Laplace mechanism with a MCMC burn-in of $x \in \{10,000; 30,000; 50,000\}$ (Reimherr et al., 2021), and *TanG-Analytic* the proposed tangent Gaussian mechanism (analytic version). (b) Utility with varying burn-ins for *Rie-Laplace*. Plots (a, b) use different matrix sizes $k$. Plot (c) explores if the bound from Th. 5 is tight in practice.

# B  Experiments

All experiments were run on DELL XPS 17 9710 LAPTOP which has 32GB OF RAM, 11TH GEN INTEL(R) CORE I9-11900H @ 2.50GHz Processor. No GPUs were used in the experiments.

## B.1  Implementation Details

Let $k \in \mathbb{N}$, $M \in \mathrm{SPD}(k)$, $\sigma > 0$ and $\rho_{\mathrm{LE}}$ denote log-Euclidean distance. The Riemannian Laplace distribution with log-Euclidean distance is given by

$$p(X|M,\sigma) = \frac{1}{\mathcal{C}_{M,\sigma}} \exp\left(-\frac{\rho_{\mathrm{LE}}(X,M)}{\sigma}\right). \tag{25}$$

Note that sampling from Eq. equation 25 requires Markov Chain Monte Carlo (MCMC) methods (Robert et al., 1999), for which one needs to choose a proposal distribution that generates candidates on the SPD Manifold. We choose the Log Gaussian distribution as the proposal in our experiments given its simplicity and the fact that it is quick to sample from. In all experiments, we found that using the log Gaussian distribution as proposal yields a stable acceptance ratio of 50% to 65%. To summarize,

1. Initialize $X_{\mathrm{curr}}$ at a random point of the manifold $\mathrm{SPD}(k)$.

2. For $1 \rightarrow n$ iterations
    (a) Draw a candidate from $X \sim \mathcal{LN}(X_{\mathrm{curr}}, \sigma^2 I)$.
    (b) With probability $\exp(-\rho_{\mathrm{LE}}(X_{\mathrm{mean}}, X)/\sigma)/\exp(-\rho_{\mathrm{LE}}(X_{\mathrm{curr}}, X)/\sigma)$ accept the generated candidate $X$ and set $X_{\mathrm{curr}} = X$ .

The final sample is chosen based on a burn-in period of 50,000 steps. Both Riemannian Laplace and tangent Gaussian mechanism can be easily implemented using existing libraries like geomstats Miolane et al. (2020), tensorflow-riemopt Miolane et al. (2020); Smirnov (2021), rieoptax Utpala et al.. In all our experiments we used geomstats Miolane et al. (2020).

## B.2  Additional Experiments

We compare the times required to privatize the Fréchet mean using both mechanisms and varying $k \in \{2, 5, 10, 15, 20, 25, 30\}$ in Fig. 4(a). Note that we used MCMC for Riemannian Laplace and its time depends on the burn-in - that we choose in $\{10000, 30000, 50000\}$. For $k = 30$, Fig. 4(a) shows that Riemannian Laplace mechanism takes 14 sec (burn-in 10000), 36 sec (burn-in 30000) and 73 sec (burn-in 50000) - whereas

our tangent Gaussian (Analytic) mechanism takes 1.3 *microsec*. Fig. 4(*b*) considers the effect of the burn-in on the Riemannian Laplace's utility and finds no significant difference for burn-ins in $\{10000, 30000, 50000\}$.

Fig. 4 (*c*) shows that the bound derived in Th. 5 is tight in practice, as illustrated by the ratio of the bound obtained in Th.5 and the practical bound.

### B.3  Code

Code is attached with Supplementary Material.

