# OpenReview forum: "Differentially Private Fréchet Mean on the Manifold of Symmetric Positive Definite (SPD) Matrices with log-Euclidean Metric"
_TMLR — Accepted by TMLR_

### Review · Reviewer_NgEb · 2022-11-18

**Summary Of Contributions:**

This paper studies the problem of differentially privacy for geometric statistics and learning. This work proposed a tangent Gaussian mechanism for private Fréchet means on the manifold of symmetric positive definite (SPD) matrices endowed with log-Euclidean metric.

The main contributions of the paper:
1. This paper proposes a tangent Gaussian Mechanism which privatizes any statistical summary on the manifold of Symmetric Positive Definite (SPD) matrices endowed with the log-Euclidean metric. The authors prove that it achieves approximate differential privacy.

2. When the statistical summary is the Fréchet mean, the authors show that the presented mechanism obtains significant improvement in terms of utility over recent works. The mechanism is computationally simple, does not rely on additional parameters, and is multiple orders of magnitude faster than the state-of-the-art.

3. The mechanism on synthetic and real-world (medical) imaging data shows effective and the authors prove a theoretical bound on the radius of log-Euclidean geodesic ball with the covariance descriptor pipeline.



**Audience:**

Yes

**Broader Impact Concerns:**

There are no any concerns on the ethical implications of the work that would require adding a Broader Impact Statement.

**Claims And Evidence:**

Yes

**Requested Changes:**

The method in the paper seems not novel. I think it can be seen as only a natural extension of the method proposed in Reimherr et al. (2021). I think the authors should emphasize the novelty their work and difference between their work and Reimherr et al.



**Strengths And Weaknesses:**

Developing a toolbox for DP on manifolds is an interesting problem. Symmetric positive definite (SPD) matrices model a wide range of data, such as medical data, thus private statistical computations on the SPD manifold are a worthy endeavour. Despite its importance for the processing of a number of medical data, geometric statistics currently stands understudied from the lens of differential privacy.

The strengths of the paper are as follows.
1. This paper proposed a tangent Gaussian mechanism that is specific to the SPD manifold equipped with the log-Euclidean metric, and the authors prove that it achieves approximate differential privacy.

2. When the statistical summary is the Fréchet mean, the mechanism outperforms the only existing baseline and is computationally simple, does not rely on additional parameters.

The weaknesses of the paper:
The proposed mechanism is restricted to one manifold with one specific metric, while the log-Euclidean metric is one of the most important metrics on the SPD manifold.

---

> ### Author Response · Authors · 2022-12-07
> **Response**
>
> We thank the reviewer for the review.
>
>
> **Differences between Riemannian Laplace [1] and Tangent Gaussian (ours)**
>
> **A** [1] and our work are completely different in many ways. Specifically
>
> *  [1] provides a Pure DP mechanism, whereas ours is an Approximate DP mechanism. [1] uses Riemannian Laplace given in [2]  and our mechanism uses log-Gaussian distribution [3]. Approximate DP mechanisms utility has better dependence on dimension. Further, our method doesn't require MCMC sampling. Hence our method is significantly better than [1] in terms of both utility and speed for higher dimensions. That being said, our method is only applicable to SPD matrices (most commonly used manifold in practice ) with log-Euclidean metric whereas [1] is applicable to general manifolds.
>
>
> * We derive the radius of log-Euclidean geodesic ball for covariance descriptors. This is crucial for privatizing in practice.
>
> **Refs**
>
> [1]  Matthew Reimherr, Karthik Bharath, Carlos Soto. Differential Privacy on Riemannian Manifolds
>
> [2]  Hatem Hajr, Ioana Ilea,  Salem Said, Lionel Bombrun, Yannick Berthoumieu.  Riemannian Laplace Distribution on the Space of Symmetric Positive Definite Matrices
>
> [3] Armin Schwartzman. Lognormal Distributions and Geometric Averages of Symmetric Positive Definite Matrices.

---

### Review · Reviewer_2LiX · 2022-11-29

**Summary Of Contributions:**

Motivated by both the growing use of the SPD manifold to model data as well as the need for differentially private machine learning algorithms, the paper proposes a fast differentially private Fréchet mean computation over the manifold of SPD matrices endowed with the Log-Euclidean metric. More specifically the paper proposes the "Tangent Gaussian Mechanism", which privatizes any statistical summary on the manifold of SPD matrices endowed with the log-Euclidean metric. When compared with previous work focused on the Fréchet mean, the proposed approach achieves higher utility and is considerably faster. Experiments on both synthetic and real-world data demonstrate the efficacy of the method.

**Audience:**

Yes

**Broader Impact Concerns:**

I have no concerns regarding the ethical implications of the work.

**Claims And Evidence:**

Yes

**Requested Changes:**

The paper writing is quite good already and the presentation, including figures, is clean. A couple of minor corrections are given below:

Page 7: "$\Delta \leq (2 * \sqrt{k})/(4 * 500)$" -> "$\Delta \leq (2 \cdot \sqrt{k})/(4 \cdot 500)$" (please use "\cdot" instead of "*")

Pages 9 and 11: Although it is understandable, please use better formatting than "(avg - 2 * std, avg + 2 * std)" for the 4 standard deviation band. Something like "the $(\mu-2\sigma, \mu+2\sigma)$ band is shown, where $\mu$ and $\sigma$ are the mean and standard deviation, respectively, of the associated result" would suffice.

**Strengths And Weaknesses:**

### Strengths

1. The idea of special casing to the SPD manifold endowed with the log-Euclidean metric has real utility, since the speed-up with respect to prior work is quite considerable and this is still a use case seen in practice.

2. The writing in this paper was for the most part very good and the presentation of the background material and the idea was very clean.

3. The experiments are reasonably solid, covering a range of synthetic and real world datasets.

### Weaknesses
1. It should be well noted that the SPD manifold endowed with log-Euclidean structure is flat (i.e. geometrically it is as trivial as Euclidean space). Hence, it is very unsurprising that for this specific case the introduced mechanism is far faster than the much more general mechanism given in [38] (which applies to all complete Riemannian manifolds). The novelty of the paper is thus limited, as it needs only to lift the corresponding differential privacy result in the Euclidean setting to the SPD case. It does so using a variety of existing SPD constructs (e.g. log-Gaussian distribution on SPD(k) from [39]). The privacy guarantee can be lifted out of past Euclidean literature.

### Verdict
As it stands, the paper does have some real-world utility. It presents a fast differential privatization method for any statistical summary on the SPD manifold endowed with log-Euclidean structure. The experiments demonstrate the method's utility for a number of real and synthetic datasets. Although I have some issues with the degree of novelty (as mentioned in the weaknesses section above), I think the paper (1) makes claims supported by accurate evidence and (2) would be interesting to at least some individuals in TMLR's audience, thereby meeting both criteria for acceptance. Therefore I recommend acceptance.

---

> ### Author Response · Authors · 2022-12-07
> **Response**
>
> Thank you for the positive review. We have now corrected the style issues that the reviewer pointed out.

---

### Review · Reviewer_Shyk · 2022-12-04

**Summary Of Contributions:**

This paper studies the problem of releasing a differentially private statistical query for data points that live in a Riemannian manifold. Specifically, it presents an extension of the well-known Gaussian mechanism to privately compute the Fréchet mean of a set of points living in the manifold of symmetric positive definite matrices. After presenting this new scheme, the article provides several results demonstrating the privacy and utility guarantees of the method. Finally, it presents a comparison of the numerical performance of the newly proposed Gaussian mechanism with the state-of-the-art solution for differentially private estimation of the Fréchet mean (i.e. the Riemannian Laplace mechanism). Based on experimental results (on synthetic and benchmark datasets), the proposed technique is claimed to outperform the Laplace mechanism both in terms of utility and computational efficiency.

**Audience:**

Yes

**Broader Impact Concerns:**

No concern

**Claims And Evidence:**

Yes

**Requested Changes:**

For securing my recommendation for acceptance, I would ask to respond to my concerns regarding the 'unfair' comparison with [1] and to clarify the technical contributions of the paper.  Also,I would suggest providing a concrete proof for Theorem 2. To be honest, I am quite convinced that following the exact same steps as previous proofs would give the expected result, but as the paper is about manifolds, there might be some technical difficulties I am not seeing. Anyway, having a concrete explanation of why the results readily apply would help.

**Strengths And Weaknesses:**

### Strengths

- *Motivation.* The focus of the article is clear and well motivated. The problem of designing methods to provide accurate and privacy-preserving statistical summaries of sensitive data is a very relevant topic in computer science, especially when considering medical applications (as mentioned in the paper). Moreover, Section 1.1. makes a good case that the tools of Riemannian geometry can naturally be used in several highly sensitive medical applications. Hence the need to implement privacy preserving mechanisms for data from Riemannian geometry.

- *Soundness.* The proposed solution is a simple adaptation of an existing method in the differential privacy literature, but the adaptation seems technically sound (at least to my level of reading of the proofs). In addition, the proposed method is empirically tested on many synthetic and real-world datasets, including some datasets related to medical applications (which I really appreciated).

- *Readability.* Overall, the paper is quite well written, easy to follow and well organized.

### Weaknesses

- *Hard to identify technical contributions.* The way the results are presented in the article gave me the impression that each result is an independent technical contribution. However, as I understand it, some of them are more important than others. Specifically, it seems that Theorems 1 and 4 represent the main technical contribution of the article.  In fact, Theorem 3 seems to me a direct corollary of [1, Theorem 2] when $\kappa \leq 0$ and Theorem 2 follows the steps of existing proofs in the differential privacy literature (as explained in the appendix). I think that highlighting this fact would improve the clarity of the paper's contributions.

- *`Unfair' comparison to [1].* The performance claims of the paper are mostly based on a comparison with the Riemannian Laplace mechanism introduced in [1]. First of all, I am not very comfortable with this comparison because, in essence, the two mechanisms do not serve the same purpose. Indeed, the Laplace mechanism aims at providing *pure* differential privacy while the Gaussian mechanism aims at providing *approximate* differential privacy. In fact, *pure* differential privacy is generally considered to be a much stronger privacy primitive than its approximate counterpart, which essentially leads to worse performance compared to approximate differential privacy. I am not saying that this comparison is a deal breaker, but I would advise relying on it much less when making performance claims. Below I give two suggestions that could mitigate this problem.
  -  *From a theoretical point of view.* Below Theorem 4, I would extend the paragraph entitled 'Theoretical results' to compare more thoroughly the obtained results with those of [1]. In particular, I would insist on the fact that approximate and pure differential privacy do not have the same semantics and that the comparison in terms of performance should be taken with a grain of salt.
  -  *From the emprical point of view.* I think that the comparison with the Laplace mechanism can remain in the experiments, but similar to what was done in [1] for the Laplace mechanism, I would complement them by adding a comparison with an extrinsic approach.

- *Minor comments.* Here, I simply point out a few typos to be corrected and small ambiguities to be cleared up.
  -  Second line below Equation (1). 'provide**s** an upper bound for the expectation'
  -  Above Table 2: 'Arsigny et al. (Arsigny et al. 2007)'
  -  Second line of Section 4: ' that needs to **be** privatized'
  -  Algorithm 1: I think the first line can be safely removed, all the information is already in the algorithm name and in 'inputs'. Also in the same algorithm, the subroutines 'CLASSIC' and 'ANALYTIC' are used without having been introduced before.
  -  Last sentence of the first paragraph of Section 6: The sentence looks incomplete.

[1] Matthew Reimherr, Karthik Bharath, and Carlos Soto. Differential privacy over riemannian manifolds. Advances in Neural Information Processing Systems, 34, 2021

---

> ### Author Response · Authors · 2022-12-07
> **Response**
>
> We thank the reviewer for the detailed review.
>
> **Clarify the technical contributions of the paper.**
>
> **A:** We have updated the paper to reflect our contributions more clearly. Specifically, it's Theorem 1, Theorem 5.
>
> * **Theorem 1:**  We use log-Gaussian [1] distribution on SPD matrices and show that it satisfies approximate differential privacy (Theorem 2). We do this by reduction to Euclidean case and by making use of the following key observation : $\textup{vecd}(\textup{Logm})(X)$ is an isometry between $(\textbf{SPD}(k), \langle.,. \rangle_{\textup{L2}})$ and $(\textbf{SPD}(k), \langle \rangle_{\textup{LE}})$ and density function of log-Gaussian[1] precisely makes use of $\textup{vecd}(\textup{Logm})(X).$ We believe picking appropriate distribution and exploiting isometric reduction to the Euclidean case is our main contribution. As an alternative, one can choose the Riemannian Gaussian distribution [2]. It is important to note that the underlying measure for Riemannian Gaussian distribution is the Riemannian volume measure, and showing this to satisfy approximate DP is not trivial due to the lack of readily available "tail bounds."  In addition, one must resort to MCMC sampling. Our choice of distribution avoids both difficulties and provides a simple and scalable algorithm.
>
> * **Theorem 5 :** We derive the radius of the log-Euclidean geodesic ball by a careful analysis (Theorem 5) of covariance descriptors. This is crucial for privatization in practice. We would like to highlight that [3] only conducted experiments on synthetic data.
>
> **Sensitivity of Fréchet mean can be obtained from Theorem 2 of [3]**
>
> **A:** Indeed, sensitivity (Theorem 3) can also be obtained from [Theorem 2] of [3] by setting $\kappa=0$. We provide alternative, much simpler proof because we already have closed form expression of Frechet mean.  We have now added this below Theorem 3 to avoid misunderstanding. Note that we didn't mention this as our contribution.
>
>
> **Comparison between Riemannian Laplace (Pure DP) vs tangent Gaussian (Approximate DP)**
>
> **A:**  We have now emphasized the trade off between privacy guarantee and utility to make sure there is no misunderstanding. Specifically, we have added details about it in i) Contributions, Section 1.2 ii) Pure DP vs Approx DP, Section 5.
>
> **Additional Experiments with Extrinsic approach**
>
> **A:** We have now added additional experiments comparing Extrinsic approach followed in [1] (See Figure 1 bottom). It should be noted that resultant privatized Fréchet mean is no longer a SPD matrix.
>
>
> **Details about proof about Theorem 2**
>
> **A :**  Theorem 1 shows that the privacy loss of tangent Gaussian mechanism is exactly the same as the Euclidean Gaussian mechanism with one crucial difference, $\rho_{E}^2$ (Euclidean distance) is replaced by $\rho^2_{LE}$ (log-Euclidean distance). This happens because $\textup{vecd}(\textup{Logm})(X)$ is an isometry between $(\textbf{SPD}(k), \langle \rangle_{\textup{L2}})$ and $(\textbf{SPD}(k), \langle \rangle_{\textup{LE}})$ and the density function of log-Gaussian[1] precisely makes use of $\textup{vecd}(\textup{Logm})(X).$ Theorem 2 is then about getting $\sigma$ for $\epsilon, \delta$. Note that privacy loss is always a real-valued scalar random variable, regardless of whether a mechanism is manifold-valued or not.
>
> The fact that the mechanism is manifold valued comes into play while deriving the privacy loss (Taken care by Theorem 1). Once privacy loss (which is \emph{real valued scalar Gaussian}  random variable) is derived, going from privacy loss to actual privacy guarantee would not be affected by whether a mechanism is manifold-valued or not because both (classical, analytic) proofs *entirely* rely on properties of *real valued scalar Gaussian random variables*.
>
>
> That being said, we have now highlighted key steps for both mechanisms (classical, analytic) in the manuscript to increase clarity. Thank you very much for this feedback.
>
> **Refs**
>
> [1] Armin Schwartzman. Lognormal Distributions and Geometric Averages of Symmetric Positive Definite Matrices.
>
> [2] Xavier Pennec. Intrinsic Statistics on Riemannian Manifolds: Basic Tools for Geometric Measurements
>
> [3] Matthew Reimherr, Karthik Bharath, Carlos Soto. Differential Privacy on Riemannian Manifolds

---

> > ### Comment · Reviewer_Shyk · 2022-12-14
> > **Thank you for the response**
> >
> > Thank you for addressing my concerns. I will review the changes to the paper, but I am satisfied with the answers.

---

### Author Response · Authors · 2022-12-07
**Uploaded Revised Paper**

Dear Reviewers,

Thanks for the detailed and insightful reviews. We have now uploaded the revised paper.

Regards,
Authors

---

### Decision · Action_Editors · 2023-01-04

**Recommendation:** Accept as is

**Comment:**

The response and revision provided by the authors convinced the reviewers: they all recommended acceptance without any additional requested changes. Therefore, I congratulate the authors for their work and recommend this paper to be accepted as is.

**Audience:**

The topic of the paper, namely privacy-preserving computations on manifold structured data, is definitely of interest to researchers in TMLR's audience working on data geometry and/or privacy. The proposed approach also has some potential applications in privacy-aware medical imaging, as illustrated by the experiments.

**Claims And Evidence:**

The paper presents an adaptation of the Gaussian mechanism to release the Fréchet mean on the SPD manifold of a set of points living in the manifold of symmetric positive definite matrices. While the underlying technical tools are fairly standard, the formal privacy and utility claims are sound, and the approach is shown to be useful and effective on real-world data (notably from medical imaging).